# Dic-UCSNet: A Novel Feature Dictionary-Based Underwater Image Compressive Sensing Framework

## Abstract

The underwater image (UWI) is one of the main sources from which researchers can obtain underwater information, thus its quality directly determines the effect and accuracy of subsequent high-level tasks. Since most existing compressive sensing (CS) algorithms are for on-land images which differ greatly from UWIs, applying them to UWIs leaves much room for performance improvement. Compared to on-land images, there exists amount of similar features among different UWIs, which is caused by the fact that underwater scenes are simpler and contain fewer semantics. Different UWIs often share semantically-identical objects that have structural and feature similarities. To further improve the performance of CS by exploiting the inter-UWIs similarity, we propose a feature dictionary-based CS framework for UWIs, dubbed Dic-UCSNet. Specifically, we first construct a multi-scale discrete codebook as the underwater feature dictionary (UF-Dic), which can provide the inter-image similarity prior to underwater CS task. Subsequently, to better match the dictionary features with the input ones to improve the utilization of the dictionary features, we propose an underwater dictionary feature fusion module (UDFF-Module), which uses the underwater physical prior to transform the degradation style of the dictionary features to input ones, and then adaptively adjusts the dictionary features according to the difference map. Experimental results on three real-world UWIs datasets show that compared with other state-of-the-art CS methods, our Dic-UCSNet can achieve an average improvement of 5% to 15% in objective metrics (PSNR/SSIM/LPIPS/NIQE) and obtain the best visual quality under all testing sampling rates (0.01, 0.04, 0.1 and 0.3).

## 1 Introduction

In recent years, ocean exploration and detection have attracted more and more scholars because the ocean contains many resources to be developed, such as rich mineral, energy (Toro et al., 2020), biological (Xiong et al., 2023) and derived tourism resources (Chen et al., 2010), etc., which are of great value for the development of human beings. Due to the special oxygen-free environment, underwater images (UWIs) are one of the main sources people can obtain underwater information, and their quality directly affects the effect and accuracy of subsequent related tasks (Ahn et al., 2017). However, due to the narrow bandwidth and limited underwater acoustic transmission channel resources, how to use a lower sampling rate to acquire and reconstruct high-quality images is important for underwater research.

The Compressive Sensing (CS) theory shows that if the image signal can be sparsely represented in a certain domain, it can be subsampled and completely recovered with a sampling rate much lower than that required by the Nyquist sampling law (Donoho, 2006). Recently, some works have shown that the CS theory is more suitable for UWI acquisition and reconstruction tasks with high bandwidth and transmission channel resources requirements (Monika et al., 2021; Zhuang et al., 2023). However, most of the existing high-performance CS methods (Lohit et al., 2018; Dai et al., 2019; Shi et al., 2019; Song et al., 2023; Yang et al., 2018; Shen et al., 2022; You et al., 2021; Shen et al., 2024) are designed for on-land images, which differ greatly from UWIs, thus their performance still has a lot of room for improvement. Moreover, most of them cannot perform well and a lot of details in their reconstructed images are lost under extremely low sampling rates (e.g. 0.01, 0.04), which in turn is

commonly used for underwater application scenes. This is because these methods do not consider the serious degradation of UWIs caused by the imaging characteristics process (Hanmante & Ingle, 2018), and more information will be lost at low sampling rates, so more additional information is needed to improve reconstruction quality.

Compared to on-land images, different UWIs generally have a higher similarity to each other (Islam et al., 2020). Due to the simplicity of the underwater scene and the task, different UWIs usually contain many objects with the same semantics. Although their specific manifestations are different, their appearance, color, and structure are about similar. Figure 1(a) illustrates the similarities between the two different UWIs. Specifically, the left column depicts the results for feature matching obtained by using HOG (Dalal & Triggs, 2005). The right column shows the results for the feature clustering using the K-Means algorithm (Pedregosa et al., 2011). It can be found that these two different UWIs have a certain visual similarity and a similar feature distribution. When the sampling rate is low, introducing these strong similarity between UWIs can supplement the missing features and improve the CS reconstruction quality (**detailed analysis about our motivations are presented in Appendix A.1**).

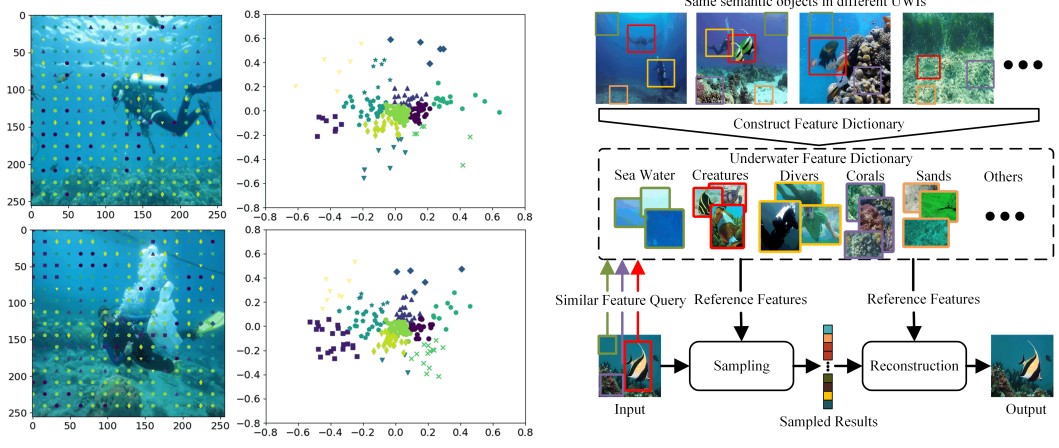

(a) Feature matching between two different UWIs       (b) Our dictionary-based CS framework

Figure 1: Motivation of our work. Different UWIs share similar texture, structure, and feature, which can be put into an underwater feature dictionary to provide reference during UWI CS task. In sub-figure (a), points with the same color and shape represent the similar features. In sub-figure (b), boxes with the same color represent objects under the same semantics.

Inspired by this characteristic, this paper proposes a novel feature dictionary-based compressive sensing framework for UWIs, dubbed Dic-UCSNet. As shown in Figure 1(b), our Dic-UCSNet uses the inter-UWIs similarity as reference information to guide the underwater CS task, reducing the sampling of similar information and providing missing information in reconstruction, so as to improve the quality of reconstruction. We first use some strictly selected UWIs (according to different water types, different semantic objects, etc.) to construct a set of discrete multiscale codebooks as the underwater feature dictionary (UF-Dic) to provide reference features. Subsequently, we design an underwater dictionary feature fusion module (UDFF-Module) to better match the reference features with the input features. As shown in Figure 2, the UDFF-Module first converts the degradation of the reference features to the input features through underwater physical priors, and then adaptively adjusts it according to the difference map with the input features. To the best of our knowledge, the proposed Dic-UCSNet is the first work to utilize inter-UWIs features as a guidance in underwater CS task. The main contributions of this paper can be summarized as follows.

- We make the first attempt to utilize the correlation of features and information among different UWIs to improve the performance of the UWI CS task and propose a novel feature dictionary-based CS framework for UWIs, which can significantly improve both objective and subjective effects at a low sampling rate.

- We construct a set of multi-scale high-quality discrete codebooks as the underwater feature dictionary (UF-Dic) to provide inter-UWIs similarity priors. The codebooks are built

according to the different underwater scenes and semantics, which can provide rich reference features. The UF-Dic can also be embedded in other end-to-end CS frameworks to improve their performance on UWIs.

- We design an underwater dictionary feature fusion module (UDFF-Module) to better match image features and dictionary features in sampling and reconstruction processes. The UDFF-Module improves the degree of matching within reference features and image features through underwater physical priors, and adaptively adjusts the feature utilization rate according to the difference map.

## 2 RELATED WORKS

### 2.1 DEEP LEARNING-BASED CS METHODS

Deep learning-based (DL-based) CS methods treat the image reconstruction problem as a learning task. Using the modeling power of neural networks, DL-based CS methods solve the time complexity problem of traditional mathematical solutions and greatly improve performance. Depending on the reconstruction network, DL-based methods can be subdivided into two categories: pure CNN-based methods and deep unfolding methods.

Pure CNN-based methods transform the CS task into an end-to-end feature mapping task. By learning the latent subsampled feature space and the corresponding feature map through training data, these methods can greatly reduce computational complexity and improve performance. DR$^2$-Net (Yao et al., 2019) and AdapRecon (Lohit et al., 2018) directly reconstruct the image from random measurements using a linear mapping network and stacked CNN layers, respectively. CSNet series (Shi et al., 2017; 2019) use residual networks to optimize the performance of feature mapping, alleviating the blocking effect generated by the two previous methods. DPA-Net (Sun et al., 2020) adds additional texture information in the reconstruction end to improve quality. However, due to the black-box nature of the neural network, this type of method is less interpretable and their performance is completely constrained by the training dataset.

Deep unfolding methods are adopted to realize traditional mathematical models. In this way, the deep unfolding method takes both performance and interpretability into account. ISTA-Net series (Zhang & Ghanem, 2018; You et al., 2021), ADMM-Net (Yang et al., 2018), and AMP-Net (Zhang et al., 2020) each translate a corresponding mathematical method into a neural network implementation. In DPC-DUN (Song et al., 2023), a dynamic path controllable depth unrolling network is used to achieve the effect of reconstruction of different depth networks for different complexity images. However, the sampling matrix used in these methods is manually set, which cannot guarantee that the key points of the image are obtained, limiting their performance to a certain extent.

### 2.2 TRANSFORMER-BASED CS METHODS

Recently, the Transformer architecture, which has achieved remarkable results in the field of NLP (Vaswani et al., 2017), has also been introduced into the image CS task. Reconstruction networks based on the swin transformer (Liu et al., 2021) have been widely proposed. TCS-Net (Gan et al., 2023) is the first purely transformer-based method that employs a block-to-pixel transformer architecture to fine-tune image reconstruction, achieving high-quality image restoration. TransCS (Shen et al., 2022) and MTC-CSNet (Shen et al., 2024) combine the advantages of Transformer and CNN to design an effective hybrid architecture for image CS. However, these methods only consider the internal features of a single image, and their performance still has a lot of room for improvement when applied to UWIs.

### 2.3 CS METHODS FOR UNDERWATER IMAGES

Currently, there are few CS algorithms specifically designed for underwater optical images. Shi et al. (2015), Monika & Dhanalakshmi (2024) and Zhuang et al. (2023) introduce the characteristics of UWIs in their methods. Shi et al. (2015) and Monika & Dhanalakshmi (2024) sample and reconstruct underwater images based on frequency and energy distribution, respectively. However, they focus more on image preprocessing and do not design the CS method itself, so their performance is limited. Zhuang et al. (2023) incorporate the underwater imaging model and prior knowledge into the CS

process for the first time and achieve good performance. However, Zhuang et al. (2023) cannot perform well at extremely low sampling rates due to the large loss of structure information.

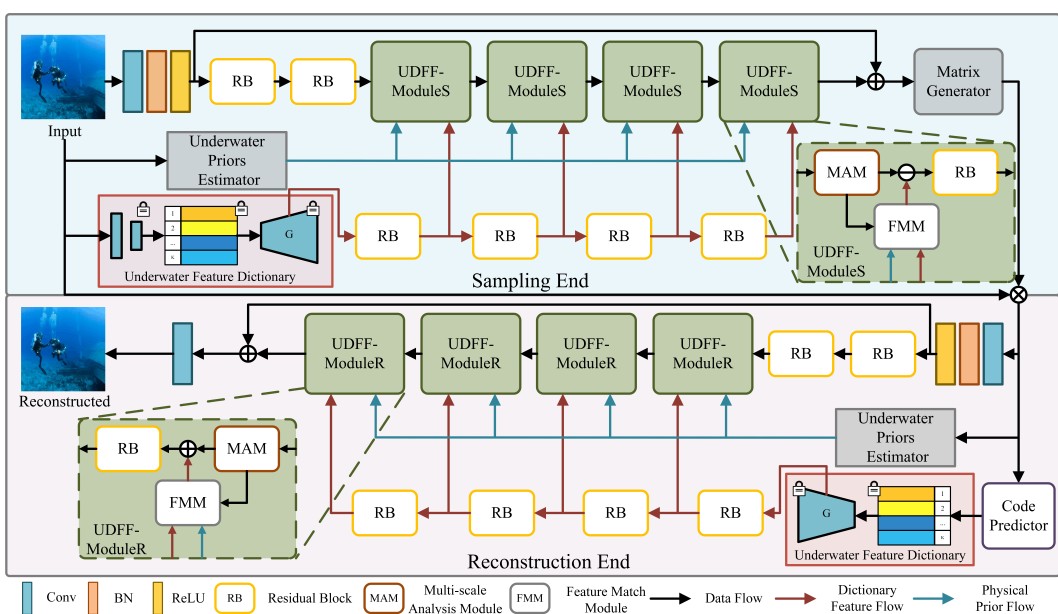

Figure 2: Architecture of our proposed Dic-UCSNet. Specifically, our Dic-UCSNet is consisted of two main important parts, the **Underwater Feature Dictionary (UF-Dic)**, which is responsible for providing inter-UWIs similarity prior; and the **Underwater Dictionary Feature Fusion Module (UDFF-Module)**, which is designed to improve the match degree between dictionary features and input features under the guidance of underwater priors.

## 3 PROPOSED METHOD

### 3.1 OVERVIEW OF THE DIC-UCSNET

As shown in Figure 2, our Dic-UCSNet contains two main important components: UF-Dic, which consists of a set of pre-trained codebook and a generator, is mainly responsible for providing reference features in the sampling and reconstruction stages, and the UDFF-Module(S/R), which is mainly used to transform dictionary features and progressively match them with image features by utilizing the underwater physical priors.

Specifically, in the sampling end, the input image is first fed into the pre-trained UF-Dic to query the reference features. Meanwhile, the underwater physical priors of the input image are estimated through the network (Lin et al., 2020). Then, the input image and the queried reference features are respectively passed through several cascaded ResBlocks (He et al., 2016) for feature extraction. Due to the high similarity between the reference features and the image features, we hope to reduce the sampling weight of the existing similar features in the dictionary and to assign more attention to the unknown features, and those unsampled features are supplemented with reference features at the reconstruction end. Therefore, in the sampling process, the reference features are subtracted from the input feature, and in the reconstruction process, the reference features are added to the image features. Finally, the corresponding sampling matrix is generated according to the adjusted features by the Matrix Generator proposed by Zhuang et al. (2023).

In the reconstruction end, the structure of the network and the data flow process are similar to those in the sampling end, except for the way of querying the feature dictionary. Because the sampled image loses a lot of texture information, it is impossible to directly query the dictionary by the same method at the sampling end. However, if the codeword queried at the sampling end is transmitted directly to the reconstruction end, it will increase the burden of the transmission channel, which does not meet the needs of practical applications. Therefore, we design a code predictor that can predict the codewords from the sampled result. Specifically, we train the code predictor by calculating

cross-entropy between the predicted codeword and the one obtained at the sampling end, so that the prediction results can be as consistent as possible with the sampling end. The structure of the code predictor and the associated ablation experiments are presented in the Appendix A.2.

## 3.2 VQGAN BASED UNDERWATER FEATURE DICTIONARY (UF-DIC)

To apply the inter-UWIs similarity to the CS task, a feature dictionary that stores UWI features is first constructed. We contrast a variety of network structures that can be used as dictionaries (**detailed experiments are reported in Appendix A.4**), and finally a modified VQGAN (Esser et al., 2021) is applied to train some selected UWIs to obtain four high-quality codebooks on different scales and a corresponding Multi-scale Fusion Generator (MSFG) as the feature dictionary, which are shared on the sampling and reconstruction ends to ensure consistency of matching features. The dictionary construction process is illustrated in the left column of Figure 3. We constrain the output of the dictionary to be as consistent as possible with the input, so that the codebooks are more apt to capture the common features in the input UWI and improve its generalization ability. Specifically, the selected UWIs are first passed through four cascaded feature extractors to obtain features from coarse to fine. Then, the nearest-neighbor matching operation is applied within four scale learnable codebooks $\mathbf{C_i}$ to produce the quantized UWI feature $\mathbf{F_i}$ and the corresponding query code sequence $\mathbf{S_i}$, in which $i \in \{1, 2, 3, 4\}$ is the corresponding scales ($128\times128$, $64\times64$, $32\times32$ and $16\times16$). This process can be expressed as follows.

$$\mathbf{F}_i = \underset{\mathbf{c}_n \in \mathbf{C_i}}{\arg\min} \|\mathbf{F}_x - \mathbf{c}_n\|, \mathbf{S}_i = \underset{n}{\arg\min} \|\mathbf{F}_x - \mathbf{c}_n\| \tag{1}$$

where $\mathbf{c}_n$ means the $n$-th feature vector in $\mathbf{C_i}$, $\mathbf{F}_x$ represents the feature of the input image. Finally, MSFG reconstructs the high-quality generated UWIs using $\mathbf{F}_i$ from each $\mathbf{C_i}$.

To make the features stored in the dictionary more extensive and representative, a part of the UWIs from the current largest real-world UWI dataset UVTD (Wang et al., 2025) which contains 5380 images is selected to train the dictionary. Moreover, to ensure that the stored features can cover the richest and most comprehensive objects and to make the number of features of each category as balanced as possible, we perform the selection of UWI in the following two steps. **1) Classification by different water types.** Since the degradation of UWI depends on the different water types (Li et al., 2019), the image obtained in clearer water has sharper textures and less blur. We first manually divide the whole UVTD into five types (blue, blue-green, green, yellow, and clear (Liu et al., 2020)) with 1999, 1646, 1143, 303 and 289 images per type. **2) Chosen according to different semantics.** Objects in underwater scenes can be roughly divided into eight categories according to their semantics (Islam et al., 2020). To ensure the balance of the number of various semantic features under each water type, we design a greedy selection algorithm and apply it to select 1/3 UWIs under each water type based on the semantic map provided in UVTD. Finally, about 1000 UWIs are chosen to construct the dictionary. The pseudo-code of this process is demonstrated in the Appendix A.3.

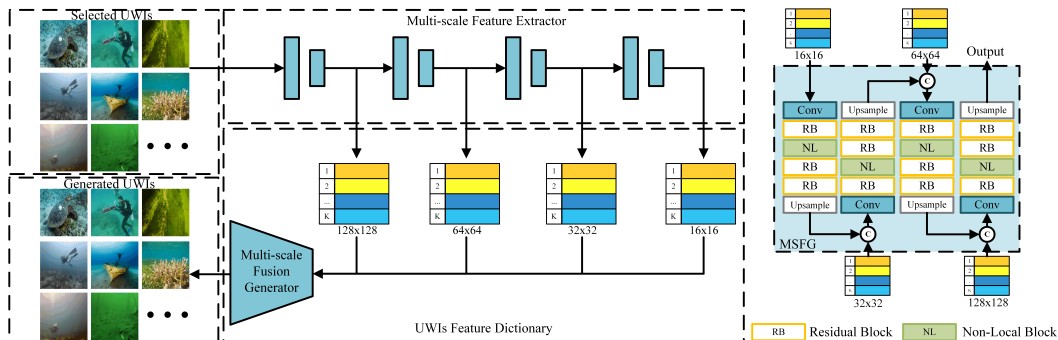

Figure 3: Construction of multi-scale UF-Dic.

MSFG is designed to better fuse the features queried from the codebooks and generate the reference features. The detailed structure of MSFG is illustrated in the right column of Figure 3. As shown, in MSFG, a generator embedded with a Non-Local Block (Wang et al., 2018) is set for each scale codebook, and the generated features on each scale are concatenated in order from small to large, which realizes the gradual improvement from local to global and texture to structure. Moreover, since

our dictionary is built from features, it can be easily trained and embedded in other end-to-end CS algorithms to improve their performance, as shown in Section 4.2.

### 3.3 UNDERWATER DICTIONARY FEATURE FUSION MODULE (UDFF-MODULE)

Maximizing the utilization of reference features is the key to determining the performance of our proposed Dic-UCSNet. For more accurate matching and better utilization of reference features, the UDFF-Module is proposed to modify the reference features, eliminating the degradation difference between the reference and input features, only retaining the structural difference, to improve the accuracy of feature matching. Meanwhile, UDFF-Module can improve the efficiency of feature utilization by learning the difference between dictionary and input features, generating a difference map, and adaptively assigning weights to reference features according to the difference map. As illustrated in Figure 4, the input feature $F_{in}$ is first analyzed in the Multi-scale Analysis Module (MAM). Meanwhile, the reference feature is transformed into DFVM. Finally, the two features are fused according to the difference map calculated between them and generate the fined feature $F_{out}$.

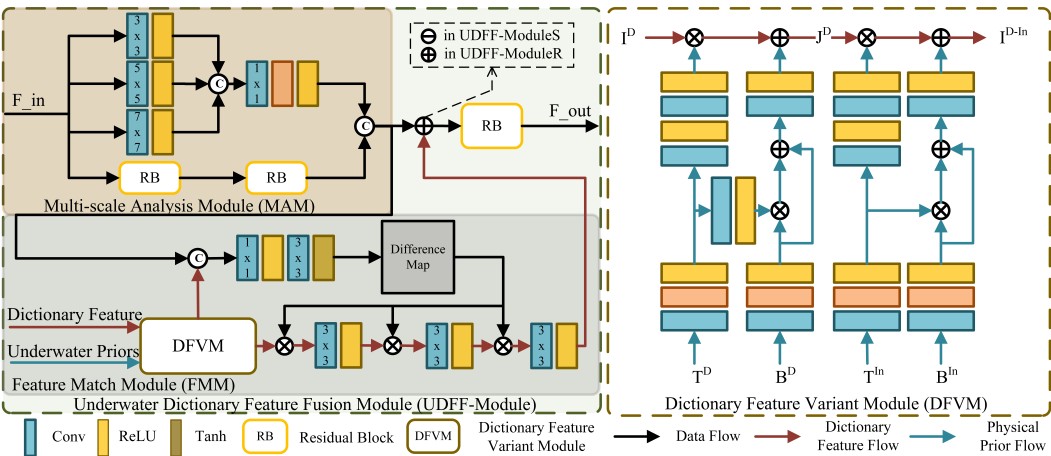

Figure 4: Overview of the proposed UDFF-Module.

**Multi-scale Analysis Module (MAM).** There are long-distance feature correlations in UWIs, and similar features may be distributed at distant locations (Zhuang et al., 2023). Convolutions with a single receptive field cannot adequately capture this long-range feature correlation. Therefore, to analyze the features more comprehensively, MAM is proposed. It consists of two parallel feature analysis branches: one is made up of a set of convolutional layers with different receptive fields, aiming to capture global features, and the other contains two cascaded ResBlocks, aiming to analyze local features. Through this multiscale analysis, the input features can be better deconstructed and the feature extraction can be more comprehensive.

**Feature Match Module (FMM).** Since the reference features are generated from a set of pre-trained codebooks, they are the concatenation of many similar features that are different from the input features. It is embodied in the following aspects: 1) the underwater degradation of the reference feature is inconsistent with the input feature, which makes the reference feature cannot be utilized fully. 2) The location and form of the reference feature are different from the input feature, which means that the reference feature cannot be used accurately. To improve the utilization efficiency of reference features, the FMM is proposed to adaptively adjust the reference feature according to the difference between the reference feature and the input feature. Firstly, the embedded DFVM normalizes the degradation of the reference feature and the input feature according to their underwater physical prior, which can greatly improve the matching accuracy. Subsequently, a difference map is calculated between the fined reference features and the input features, which further quantifies the differences. The difference map is then used as a weight controller to adaptively adjust the reference features. Higher weights will be allocated for parts with large differences, thus improving the accuracy of the usage of reference features.

**Dictionary Feature Variant Module (DFVM).** DFVM is proposed to transform the degradation of reference features to that of input features. By analyzing the underwater imaging process, it can

be found that underwater imaging is mainly affected by two physical priors, background light and transmission map, which can be expressed as the following equation (Li et al., 2017).

$$I(x) = J(x)e^{-\beta d(x)} + B(x)\left(1 - e^{-\beta d(x)}\right) = J(x)T(x) + B(x)(1 - T(x)) \tag{2}$$

where $x$ is the index of pixels in the image. $I(x)$ is the observed UWI, $J(x)$ is the clear image with no degradation. $B(x)$ and $T(x)$ are background light and transmission map, which indicate global color tone and the distance from camera to objects, respectively (the visualization examples are provided in the Appendix A.7). Based on Eq. 2, the degradation of dictionary features can be transformed to input features according to the following.

$$
\begin{aligned}
I^{D-In}(x) &= J^D(x)T^{In}(x) + B^{In}(x)(1 - T^{In}(x)) \\
&= \left[\frac{1}{T^D(x)}I^D(x) - \frac{B^D(x)}{T^D(x)} + B^D(x)\right]T^{In}(x) + B^{In}(x)(1 - T^{In}(x))
\end{aligned} \tag{3}
$$

where $D$ and $In$ denote the corresponding elements of the dictionary and input features, respectively.

Based on the above analysis, the DFVM is designed to parametrically implement Eq. 3. The detailed structure of DFVM is presented in the right column of Figure 4, $I^D$ is first transformed to $J^D$ by its own physical priors, and then transformed to $I^{D-In}$ by the prior of the input features. The network structure of DFVM is designed strictly according to Eq. 3, and the normalization of degradation can be achieved through end-to-end training.

## 4 EXPERIMENTAL RESULTS

In this section, extensive experiments are conducted to verify the superiority of our Dic-UCSNet. First, quantitative and qualitative comparisons are performed on three real-world underwater datasets UVTD-T(Wang et al., 2025), UIEB (Li et al., 2019) and SUIM (Islam et al., 2020) with other SOTA methods, including DL-based methods (Dai et al., 2019; Shi et al., 2019; You et al., 2021; Song et al., 2023), transformer-based methods Shen et al. (2022; 2024) and the underwater method (Zhuang et al., 2023) to demonstrate the performance and generalizability of Dic-UCSNet. Then, a series of ablation studies are conducted to verify the contribution of each proposed module. The details of experiment settings are illustrated in the Appendix A.2.

Table 1: Performance Comparisons on three Real-World Underwater Datasets.

| Dataset | Methods | SR = 0.01 | | | | SR = 0.04 | | | | SR = 0.1 | | | | SR = 0.3 | | | |
|---|---|---|---|---|---|---|---|---|---|---|---|---|---|---|---|---|---|
| | | PSNR↑ | SSIM↑ | LPIPS↓ | NIQE↓ | PSNR↑ | SSIM↑ | LPIPS↓ | NIQE↓ | PSNR↑ | SSIM↑ | LPIPS↓ | NIQE↓ | PSNR↑ | SSIM↑ | LPIPS↓ | NIQE↓ |
| UVTD | AdapRecon | 20.38 | 0.492 | 0.701 | 15.10 | 23.52 | 0.606 | 0.546 | 12.41 | 25.71 | 0.708 | 0.442 | 9.89 | 29.12 | 0.816 | 0.218 | 8.94 |
| | CSNet⁺ | 23.31 | 0.603 | 0.527 | 12.15 | 25.07 | 0.701 | 0.414 | 10.69 | 26.32 | 0.781 | 0.375 | 9.65 | 29.69 | 0.880 | 0.211 | 6.36 |
| | ISTA-Net⁺ | 23.36 | 0.589 | 0.517 | 11.16 | 24.85 | 0.682 | 0.418 | 10.42 | 26.95 | 0.788 | 0.354 | 8.99 | 30.12 | 0.916 | 0.188 | 5.92 |
| | DPC-DUN | 18.97 | 0.462 | 0.883 | 17.07 | 27.62 | 0.719 | 0.332 | 9.55 | 31.21 | 0.818 | 0.266 | 7.15 | 36.64 | 0.932 | 0.093 | 6.19 |
| | TransCS | 24.06 | 0.597 | 0.503 | 10.26 | 25.99 | 0.724 | 0.398 | 9.15 | 27.73 | 0.882 | 0.306 | 8.42 | 30.52 | 0.911 | 0.177 | 6.04 |
| | MTC-CSNet | 24.22 | 0.575 | 0.509 | 10.87 | 26.38 | 0.772 | 0.376 | 8.92 | 28.15 | 0.860 | 0.285 | 8.61 | 30.87 | 0.904 | 0.180 | 5.97 |
| | UCSNet | 25.89 | 0.683 | 0.402 | 7.95 | 30.18 | 0.826 | 0.243 | 8.34 | 33.19 | 0.894 | 0.152 | 7.07 | 36.14 | 0.949 | 0.071 | 5.40 |
| | Ours | 27.02 | 0.732 | 0.315 | 6.891 | 31.13 | 0.840 | 0.203 | 6.47 | 33.58 | 0.922 | 0.120 | 5.61 | 36.88 | 0.961 | 0.065 | 5.08 |
| UIEB | AdapRecon | 22.55 | 0.548 | 0.613 | 12.19 | 26.11 | 0.671 | 0.449 | 9.34 | 28.51 | 0.804 | 0.399 | 8.67 | 33.25 | 0.896 | 0.213 | 7.66 |
| | CSNet⁺ | 24.57 | 0.623 | 0.525 | 11.34 | 26.77 | 0.728 | 0.399 | 8.67 | 28.62 | 0.816 | 0.336 | 8.44 | 33.56 | 0.909 | 0.174 | 7.42 |
| | ISTA-Net⁺ | 23.94 | 0.596 | 0.515 | 10.19 | 26.22 | 0.696 | 0.423 | 8.71 | 28.48 | 0.806 | 0.356 | 8.34 | 33.16 | 0.907 | 0.185 | 7.59 |
| | DPC-DUN | 18.63 | 0.444 | 0.795 | 17.23 | 27.70 | 0.724 | 0.347 | 9.76 | 31.55 | 0.833 | 0.226 | 5.75 | 37.22 | 0.941 | 0.095 | 5.75 |
| | TransCS | 24.52 | 0.628 | 0.525 | 9.49 | 27.18 | 0.761 | 0.370 | 8.31 | 29.42 | 0.828 | 0.309 | 8.02 | 33.88 | 0.944 | 0.131 | 6.80 |
| | MTC-CSNet | 24.98 | 0.671 | 0.529 | 9.07 | 27.35 | 0.764 | 0.354 | 8.89 | 29.92 | 0.878 | 0.303 | 8.12 | 33.51 | 0.954 | 0.134 | 6.22 |
| | UCSNet | 26.09 | 0.696 | 0.408 | 7.74 | 30.47 | 0.836 | 0.251 | 8.44 | 33.57 | 0.902 | 0.156 | 6.62 | 36.50 | 0.963 | 0.075 | 5.02 |
| | Ours | 27.27 | 0.749 | 0.321 | 6.13 | 31.58 | 0.860 | 0.210 | 5.29 | 34.24 | 0.913 | 0.124 | 5.29 | 37.31 | 0.975 | 0.061 | 4.88 |
| SUIM | AdapRecon | 20.28 | 0.483 | 0.724 | 14.01 | 23.30 | 0.594 | 0.569 | 10.12 | 25.60 | 0.749 | 0.458 | 8.85 | 32.95 | 0.867 | 0.194 | 7.86 |
| | CSNet⁺ | 22.19 | 0.552 | 0.652 | 13.01 | 24.02 | 0.661 | 0.431 | 9.55 | 27.49 | 0.761 | 0.340 | 8.51 | 33.06 | 0.874 | 0.191 | 7.45 |
| | ISTA-Net⁺ | 21.85 | 0.528 | 0.664 | 13.39 | 23.85 | 0.634 | 0.437 | 9.83 | 27.07 | 0.754 | 0.387 | 8.55 | 33.05 | 0.873 | 0.188 | 7.58 |
| | DPC-DUN | 18.24 | 0.428 | 0.789 | 17.05 | 25.58 | 0.679 | 0.345 | 8.70 | 28.88 | 0.790 | 0.235 | 6.75 | 34.19 | 0.920 | 0.104 | 5.89 |
| | TransCS | 23.17 | 0.580 | 0.542 | 12.30 | 24.72 | 0.706 | 0.332 | 9.28 | 28.40 | 0.757 | 0.332 | 7.17 | 33.38 | 0.923 | 0.145 | 6.81 |
| | MTC-CSNet | 23.76 | 0.594 | 0.535 | 12.79 | 24.79 | 0.714 | 0.350 | 9.07 | 28.95 | 0.768 | 0.319 | 7.87 | 33.67 | 0.933 | 0.143 | 6.27 |
| | UCSNet | 23.92 | 0.635 | 0.438 | 7.86 | 28.07 | 0.791 | 0.271 | 8.13 | 31.11 | 0.875 | 0.168 | 6.80 | 34.16 | 0.943 | 0.074 | 5.02 |
| | Ours | 25.30 | 0.703 | 0.326 | 6.59 | 29.33 | 0.831 | 0.214 | 6.25 | 32.19 | 0.900 | 0.126 | 5.32 | 35.21 | 0.966 | 0.051 | 4.12 |

The **best** and second-best results are **bolded** and underlined respectively.

### 4.1 COMPARISON WITH STATE-OF-THE-ART CS METHODS

We conduct quantitative and qualitative comparisons with seven SOTA algorithms (including two pure CNN methods: CSNet⁺ (Shi et al., 2019) and AdapRecon (Dai et al., 2019), two deep unfolding methods: ISTA-Net⁺ (You et al., 2021) and DPC-DUN (Song et al., 2023), two transformer-based methods: TransCS (Shen et al., 2022) and MTC-CSNet (Shen et al., 2024), and one underwater method UCSNet (Zhuang et al., 2023)), on UVTD-T (Wang et al., 2025), UIEB (Li et al., 2019), and SUIM (Islam et al., 2020) datasets, which contain 876, 950, and 1525 images, respectively, to

evaluate performance and generalizability. For quantitative comparison, the pixel and feature-level metrics PSNR and the Learned Perceptual Image Patch Similarity (LPIPS) (Zhang et al., 2018), the human visual perception level metrics SSIM, and the Natural Image Quality Evaluator (NIQE) (Mittal et al., 2012) are used for evaluation. As listed in Table 1, our Dic-UCSNet achieves the best performance on all testing datasets at every sampling rate. Especially when the sampling rate is extremely low (e.g. 0.01, 0.04), our Dic-UCSNet exhibits excellent performance and outstanding generalization ability. For example, our Dic-UCSNet can achieve at least 1.18 dB/0.063/0.087/1.04 gains in PSNR/SSIM/LPIPS/NIQE on UIEB at a sampling rate of 0.01. Because under the low sampling rate, a lot of information is lost, but this lacking can be made up for by introducing the inter-UWIs information through the dictionary as the reference for the reconstruction network.

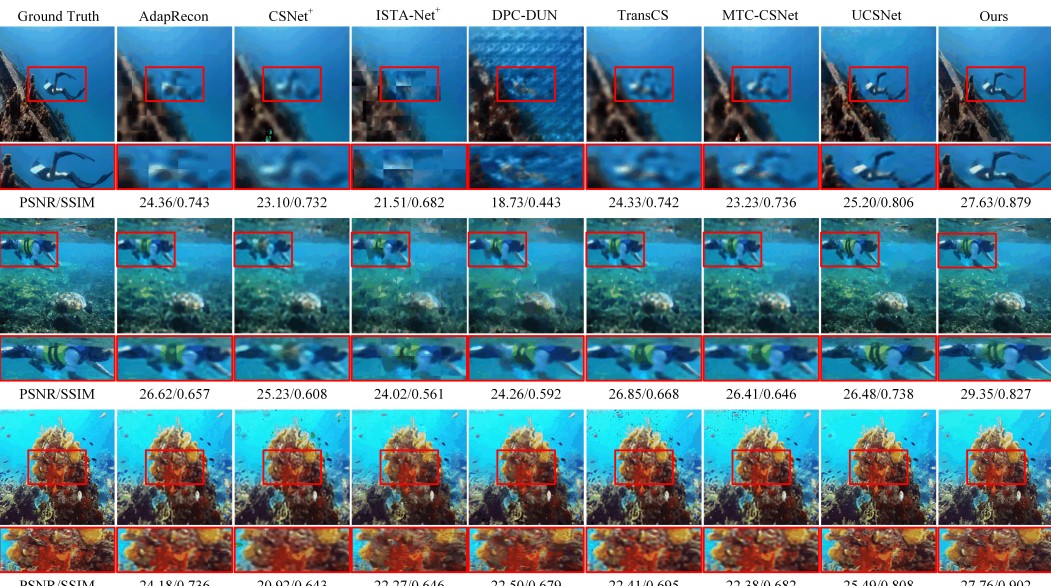

Figure 5: Visualization of qualitative comparisons. Row 1: examples from UVTD dataset, at the sampling rate of 0.01; Row 2: examples from UIEB dataset, at the sampling rate of 0.04; and Row 3: examples from SUIM dataset, at the sampling rate of 0.1.

For qualitative comparison, some results from three testing datasets under three sampling rates are visualized (we only present the samples under the sampling rates of 0.01, 0.04 and 0.1, since the reconstruction results of all methods are very good (PSNR¿30 dB) at the sampling rate of 0.3, the subjective differences are not obvious.). As shown in Figure 5, our Dic-UCSNet can achieve the best restoration in both overall visual quality and details. Especially for extreme low sampling rates (0.01, 0.04), our method can better restore the detail texture, and the objects in the image can be distinguished easily, while other methods have certain visual defects, such as blocking effect, edge artifacts, detail blurry, etc. More visualization results are also provided in the Appendix A.9.

## 4.2 Ablation Studies

In this subsection, we mainly present the ablation study of the proposed modules and verify the gains that dictionaries bring to some other end-to-end frameworks. More ablation studies are provided in the Appendix A.5, including the UF-Dic correlation ablation studies and the robustness and generalization ability experiments of Dic-UCSNet.

**Effectiveness of the Proposed Modules.** To verify the effectiveness of the proposed FMM and DFVM in the UDFF-Module, ablation experiments are performed on the UVTD-T dataset at a sampling rate of 0.1. We demonstrate the gain brought by the proposed modules by comparing the PSNR and SSIM values of the reconstructed images in the case of different combinations of FMM and DFVM (BL is the baseline which is composed of several ResBlocks). The experimental results are reported in Table 2. It is clear that BL achieves the lowest PSNR/SSIM values of 33.58 dB/0.892 among all ablation models. And with FMM and DFVM alone, the performance can be improved to 31.18 dB/0.851 and 32.37 dB/0.869, respectively, which are quite significant improvements. It can also be seen from

Figure 6(a), both FMM and DFVM bring improvements in the details of the reconstruction images, making the texture sharper and clearer and looking more consistent with the ground truth image.

Table 2: Ablation study of the effectiveness of proposed Modules.

| Components | Different Combinations | | | |
|---|---|---|---|---|
| BL | ✓ | ✓ | ✓ | ✓ |
| FMM | ✓ | ✓ | × | × |
| DFVM | ✓ | × | ✓ | × |
| PSNR↑ | **33.58** | 31.18 | 32.37 | 25.30 |
| SSIM↑ | **0.892** | 0.851 | 0.869 | 0.848 |

Table 3: Effectiveness and generalization of the dictionary-based framework.

| Methods | PSNR↑ | SSIM↑ |
|---|---|---|
| CSNet$^+$ | 26.32 | 0.781 |
| CSNet$^+$+UF-Dic | **27.95** | **0.844** |
| AdapRecon | 25.71 | 0.708 |
| AdapRecon+UF-Dic | **27.17** | **0.808** |

**Effectiveness of the Dictionary-based Framework.** To validate the generality of our proposed dictionary-based framework, we embed our pre-trained UF-Dic into CSNet$^+$ and AdapRecon. We compare the reconstruction quality w/o the UF-Dic at a sampling rate of 0.01 on UVTD-T. The comparison results are reported in Table 3. It is clear that with UF-Dic, the performance of CSNet$^+$ and AdapRecon can be effectively promoted by approximately 2.19 dB / 0.241 and 2.79 dB / 0.226 in PSNR / SSIM, which is a significant improvement. Figure 6(b) also demonstrates the visual quality of the reconstructed image with and without the UF-Dict. It is obvious that after embedding the dictionary, the overall visual quality of the reconstructed image is greatly improved, and the artifacts are significantly reduced. Moreover, the detail quality in the enlarged red box is significantly improved, which can be accurately recognized by the human eye.

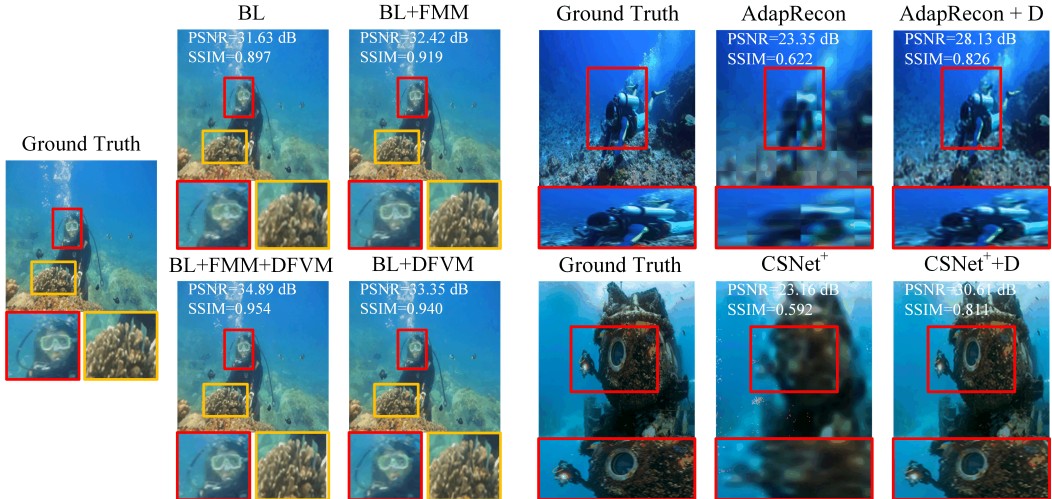

(a) Ablation on proposed modules      (b) Ablation on dictionary-based framework

Figure 6: Visualization of ablation studies on (a) proposed modules, and (b) dictionary-based framework.

## 5 CONCLUSION

In this paper, we analyze and explore how to use inter-UWI similarity to improve the performance of the underwater CS task. In this regard, a feature dictionary-based CS framework for UWIs is creatively proposed, dubbed Dic-UCSNet. We first establish a set of multi-scale codebooks from some selected UWIs as an UF-Dic to provide high-quality inter-UWIs reference features. Meanwhile, to better utilize the reference features in the sampling and reconstruction process, the UDFF-Module is proposed to normalize the degradation of the reference features to the input features under the guidance of the underwater priors, and adaptively adjust the utilization weight of the reference features according to the difference. Finally, many experiments are conducted, including performance comparisons, ablation studies of the proposed modules to prove the superiority and effectiveness of our proposed framework.

## 6 REPRODUCIBILITY STATEMENT

We claim that our Dic-UCSNet is reproducible. However, since the code of this paper is strongly relevant to our current and upcoming research, it will be open sourced upon acceptance. Meanwhile, the detailed structure of UF-Dic is demonstrated in Appendix A.11 and the implementation details are presented in Appendix A.2 for readers to reproduce the experimental results.

All the datasets mentioned and used in this paper are publicly available and can be accessed and downloaded at the URL provided in their articles. UVTD (Wang et al., 2025), UIEB (Li et al., 2019), SUIM (Islam et al., 2020).

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

# A APPENDIX

This Appendix is organized as follows:

Section A.1 presents the detailed analysis of the inter-UWIs characteristics, which is the main motivation of this paper.

Section A.2 shows the implementation details of our Dic-UCSNet and the corresponding experiment settings.

Section A.3 demonstrates the pseudo-code for selecting UF-Dic training UWIs.

Section A.4 gives the explanations about why VQGAN is selected to construct the UF-Dic.

Section A.5 provides more ablation studies, including verification of the diversity of our UF-Dic (Section A.5.1), different effect of the number of codebooks on the reconstruction quality (Section A.5.2), robustness and generalization ability studies (Section A.5.3).

Section A.6 provides the complexity comparison results of our Dic-UCSNet and other SOTA CS methods.

Section A.7 exhibits the examples of some UWIs with their corresponding underwater physical priors.

Section A.8 provides the visualization of the intermediate features of some key network components in our Dic-UCSNet.

Section A.9 demonstrates some additional visual experimental results on testing datasets under different sampling rates.

Section A.10 provides an analysis of the limitations of our Dic-UCSNet and the prospects for further future work.

Section A.11 illustrates the detailed network architecture of our Dic-UCSNet.

## A.1 INTER-UWI CHARACTERISTICS ANALYSIS

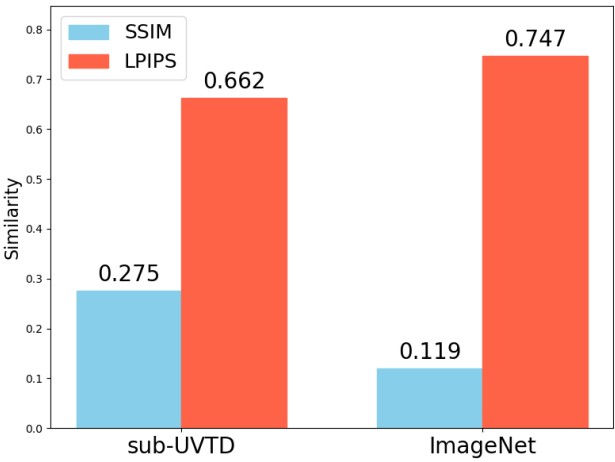

Figure 7: Similarity measurement of images in UVTD (Wang et al., 2025) and ImageNet (Russakovsky et al., 2015). Lower LPIPS and higher SSIM indicate stronger inter-image relationship.

Unlike on-land images, UWIs have relatively few semantics due to the simplicity of the underwater scenes. As shown in Figure 1(a), although the content of these two UWIs is different in the specific representation form, they have a very strong feature correlation and structural similarity, which means that the two images contain some information about each other. Consequently, UWIs captured in different water types and at varying depths can be connected through these similar features. Such inter-image correlation is not strong in on-land images. We randomly select 2000 images from UVTD ((Wang et al., 2025)) and the commonly used on-land image dataset ImageNet ((Russakovsky et al.,

2015)) and calculate their Structural Similarity Index (SSIM) and Learned Perceptual Image Patch Similarity (LPIPS) (Zhang et al., 2018) to quantify their inter-image similarity. In particular, SSIM assesses the similarity in the structure domain, while LPIPS assesses the similarity in the feature domain. For each image in the dataset, SSIM and LPIPS scores are computed against all other images (excluding itself), and the results are averaged across all comparisons. As shown by the detailed metrics in Figure 7, the mean SSIM and LPIPS for UWIs are 0.275 and 0.662, respectively, indicating stronger correlations at both the feature and structural levels compared to the on-land images. These results confirm that the similarity among UWIs is significantly higher than among on-land images. Thus, in the CS task of restoring high-quality images with a low number of samples, the reconstruction quality can be greatly improved by using known similar features to assist the reconstruction of missing ones.

## A.2 EXPERIMENT SETUPS

**Implementation Details.** Our proposed Dic-UCSNet is trained in two stages, including dictionary training and CS network training. Both stages are implemented with Pytorch and optimized by Adam on one RTX 4090D for 200 epoches. In the UF-Dic training stage, 1000 UVTD images (Wang et al., 2025) are selected to train the dictionary, the learning rates for MSFG and the discriminator are initially set to 0.0002, and they degrade 0.5 times every 50 epoches. In total, four loss functions are used to constrain the codebook and the MSFG.

$$
\begin{cases}
\mathcal{L}_{MSE} = \dfrac{1}{HW} \sum_{i=1}^{H} \sum_{j=1}^{W} |y_{(i,j)} - x_{(i,j)}|^2, \\[2ex]
\mathcal{L}_{VGG} = \dfrac{1}{C_l H_l W_l} \sum_{i=1}^{C_l} \sum_{j=1}^{H_l} \sum_{k=1}^{W_l} \left[ \mathcal{F}(y)_{(i,j,k)} - \mathcal{F}(x)_{(i,j,k)} \right]^2, \\[2ex]
\mathcal{L}_{GAN}(G, D) = \min_{G} \max_{D_1, D_2, D_3} \sum_{i=1}^{3} \mathcal{L}_{GAN}(G, D_i), \\[2ex]
\mathcal{L}_{codebook} = \|\mathrm{SG}[\mathcal{F}_h] - \mathcal{F}_c\|^2 + \alpha \|\mathcal{F}_h - \mathrm{SG}[\mathcal{F}_c]\|^2
\end{cases}
\tag{4}
$$

where the first three losses are used as image-level losses to update the MSFG. In detail, $\mathcal{L}_{MSE}$ and $\mathcal{L}_{VGG}$ ensure the consistency between the input and output images from the perspective of pixels and features, and $\mathcal{L}_{GAN}(G, D)$ is the GAN loss proposed in (Park et al., 2019). The last loss is a code-level loss, as adopted in (Esser et al., 2021; Zhou et al., 2022), which is used to reduce the distance between the quantized UWI feature $\mathbf{F_i}$ in the codebook $\mathbf{C_i}$ and the input feature $\mathcal{F}$. $\mathrm{SG}[\cdot]$ denotes the stop-gradient operator, and $\alpha$ is the weight factor to balance the update rates of the feature extractor and the codebook. Thus, the total loss for dictionary training can be expressed as follows.

$$
\mathcal{L}_{Total} = \lambda_1 \mathcal{L}_{GAN} + \lambda_2 \mathcal{L}_{MSE} + \lambda_3 \mathcal{L}_{VGG} + \lambda_4 \mathcal{L}_{codebook}
\tag{5}
$$

where $\lambda_2$, $\lambda_3$ and $\lambda_4$ are set as 0.001, 1 and 0.25 in our work. $\lambda_1$ is an adaptive value achieved from the method proposed in Esser et al. (2021).

In the training stage of CS network, another 3504 non-repeated images from UVTD (UVTD-F) are used as training data. The weight parameter of the UF-Dic is fixed during this stage. Note that at the sampling end, the input image is directly fed into the feature extractor at the dictionary training stage to match the reference features. However, in the reconstruction stage, since the sampled images lose the original structural information, the feature extractor cannot accurately match the features. As mentioned in Section 3, we design a code predictor to directly predict code from the sampled results to solve this problem. The code predictor is trained by applying a cross-entropy loss as shown in Eq. 6 with the codewords queried at the sampling end. And the structure of the code predictor is illustrated in Figure 8.

$$
\mathcal{L}_{cross-entropy} = - \sum_{i=1}^{C} y_i \log(\hat{y}_i)
\tag{6}
$$

where $C$ represents the number of codes in the codebook, $y_i$ and $\hat{y}_i$ are the query code matched at the sampling end and the query code predicted by the code predictor at the reconstruction end. In particular, in the entire training stage, all training images are resized to $256 \times 256$ and values are

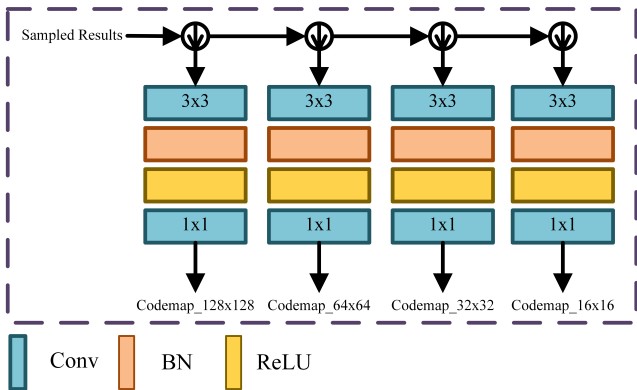

Figure 8: Overview of the designed code predictor. The sampled image is downsampled sequentially to perform codebook prediction by a combination of Conv-BN-ReLU-Conv unit at each corresponding scale.

normalized to [0, 1] to reduce the computation. Meanwhile, in the training stage of CS network, random Gaussian noise and brightness change are added to the input image to improve the robustness of the network.

Hence, the total loss for the CS network training can be expressed as follows.

$$\mathcal{L}_{Total} = \lambda_1 \mathcal{L}_{GAN} + \lambda_2 \mathcal{L}_{MSE} + \lambda_3 \mathcal{L}_{VGG} + \lambda_4 \mathcal{L}_{cross-entropy} \tag{7}$$

where $\lambda_4$ is set to 1. The other $\lambda_i$ settings and the loss function are the same as in the dictionary training stage.

We also carried out an experiment to compare the performance with/without the code predictor. Specifically, w/o code predictor means that the query codewords in the sampling end are directly used in the reconstruction end to match the reference feature from the feature dictionary. The results are reported in Table 4. Although better PSNR/SSIM performance can be achieved by directly using the query codewords from the sampling, transmitting them from the sampling end to the reconstruction end will increase the burden on the channel in practical application. However, despite the performance of using the code predictor being reduced by 0.23 dB/0.022 in PSNR/SSIM, no additional information must be transmitted in the underwater transmission channel. Considering the premise of meeting the requirements of practical applications, the performance loss is acceptable.

Table 4: Performance comparisons of w/o code predictor on UVTD under the sampling rate of 0.1.

|  | PSNR↑ | SSIM↑ | Contents needs to be transmitted |
|---|---|---|---|
| Dic-UCSNet with code predictor | 33.58 | 0.922 | sampled results + query codewords |
| Dic-UCSNet w/o code predictor | 33.81 | 0.944 | sampled results |

**Experiment Settings.** As mentioned in Section 4 in main paper, our experiments are carried out mainly on three publicly available real-world UWI datasets, including **UVTD** (Wang et al., 2025), **UIEB** (Li et al., 2019) and **SUIM** (Islam et al., 2020), which contain 5380, 950 and 1525 UWIs, respectively. Specifically, UVTD is divided into three parts, one for training feature dictionary (UVTD-D: 1000), one for training CS network (UVTD-F: 3504) and the other for testing (UVTD-T: 876). For fair comparison, all comparison methods are retrained in UVTD-F with their official code and parameter settings, whose URLs are provided in their papers (Dai et al., 2019; Shi et al., 2019; Song et al., 2023; You et al., 2021; Shen et al., 2024; 2022; Zhuang et al., 2023). Quantitative and qualitative comparisons are conducted on UVTD-T, UIEB and SUIM, and all experiments are done on the same laptop with Intel(R) Core(TM) i9-14900K, RTX 4090D and 64G memory.

A.3    PSEUDO-CODE OF GREEDY SELECTION ALGORITHM FOR DICTIONARY UWIS.

In this section, the pseudo-code of the algorithm used to select the training images for UF-Dic is demonstrated in Algorithm 1. Through this greedy selection algorithm, approximately 1000 images are chosen from UVTD. The selected images are balanced in terms of semantics and water type.

---

**Algorithm 1** Greedy Balanced Selection Algorithm

---

**Require:**
 1: images: List of (image_id, sem_labels, scene)
 2: target_size: Target number of images (approximately 1000)
 3: target_counts: Dictionary of desired counts for each sem_label and scene
**Ensure:**
 4: selected_images: List of selected images
 5: sem_label_counts: Count of each sem_label in selected images
 6: scene_counts: Count of each scene in selected images
 7: Initialize selected_images as empty list
 8: Initialize sem_label_counts as empty dictionary
 9: Initialize scene_counts as empty dictionary
10: **while** length(selected_images) < target_size **do**
11:     **for all** image (img_id, sem_labels, scene) in images **do**
12:         Calculate current label frequencies in sem_label_counts
13:         Calculate current scene frequencies in scene_counts
14:         min_label_count ← MIN(sem_label_counts.values())
15:         min_scene_count ← MIN(scene_counts.values())
16:         **if** any label in sem_labels has count == min_label_count AND scene has count == min_scene_count **then**
17:             Add img_id to selected_images
18:             **for all** label in sem_labels **do**
19:                 Update sem_label_counts for label
20:             **end for**
21:             Update scene_counts[scene] by +1
22:             **if** length(selected_images) ≥ target_size **then**
23:                 Break
24:             **end if**
25:         **end if**
26:     **end for**
27: **end while**
28: **if** length(selected_images) < target_size **then**
29:     Remaining_images ← SELECT images not in selected_images
30:     Shuffle remaining images
31:     num_needed ← target_size - length(selected_images)
32:     Add first num_needed from remaining_images to selected_images
33: **end if**
34: Return selected_images, sem_label_counts, scene_counts

---

A.4    COMPARATIVE EXPERIMENTS OF DIFFERENT DICTIONARY NETWORK STRUCTURES.

In this section, the reasons for choosing VQGAN as our underwater feature dictionary are shown. Currently, there are two main ways to build an image feature dictionary, the first is based on clustering (such as K-means), and the other is based on a codebook and a generative model (Xiaoming Li, 2020). Compared with clustering-based feature dictionary, codebook and generative model-based feature dictionary shows superiority in the diversity and flexibility of feature representation and can also be embedded into other methods by simple retraining.

We compare several current SOTA generative models (StyleGAN2 (Karras et al., 2020), Diffusion Model (Ho et al., 2020) and VQGAN (Esser et al., 2021)) from three perspectives, and finally choose VQGAN as the base model for UF-Dic. The first is the **discrete codebook mechanism**. VQGAN can generate discrete codebooks and map features into discrete vectors that can be queried

directly by vector quantization. This discreteness can cause the input images to quickly match similar features. However, the feature representation of the generative model is embedded in a continuous latent space, which requires multi-step iteration to generate specific features, so it cannot be directly used as a query-able reference feature. The second is the **structured storage ability**. By setting different scales of codebooks in VQGAN's training step, multi-scales feature can be stored. For instance, the four scales we have set can be used to capture the global, mid-level, and microscopic detail features. This design precisely matches the feature distribution of the UWIs. However, the feature hierarchy of generative models places more emphasis on style transfer than on capturing cross-image similarity, making it difficult to provide targeted reference features for UWI CS. The last is the **training difficulty and inference efficiency**. VQGAN requires a relatively lower number of training samples compared to generative models. Therefore, if the selected samples are representative, high-quality codebooks can be obtained. However, generative models require a large amount of data to achieve a good effect, which is something that currently available underwater datasets cannot provide.

To verify the above analysis, a StyleGAN2 and a Diffusion Model based dictionary are trained, respectively, with the same training data of UF-Dic, and their performance under the sampling rate of 0.1 on UVTD-T are evaluated by embedding in the same CS framework. It can be found in Table 5 that using StyleGAN2 and Diffusion Model as the dictionary results in inferior performance and longer running time compared to our dictionary based on VQGAN.

Table 5: Comparison of Different Dictionary Networks On UVTD-Tunder the sampling rate of 0.1.

| Dictionary | PSNR↑ | SSIM↑ | Inference Time (ms)↓ |
|---|---|---|---|
| StyleGAN2-based Dictionary | 31.12 | 0.881 | 51 |
| Diffusion Model-based Dictionary | 32.07 | 0.892 | 62 |
| Our VQGAN-based Dictionary | **33.58** | **0.922** | **40** |

### A.5 MORE ABLATION STUDIES

#### A.5.1 VERIFICATION OF THE DIVERSITY OF OUR UF-DIC.

As mentioned in Section 3.2 of our paper, a part of the UWIs are selected from UVTD to build the UF-Dic according to their type and semantics of water. In this part, some experiments are conducted to compare the diversity and comprehensiveness of our UF-Dic with two other dictionaries built from other UWI datasets, including UIEB (Li et al., 2019) and SUIM (Islam et al., 2020). First, the Spatial Information (SI) and Colorfulness (CF) of three sets of UWIs (UVTD-D selected by us, all UWIs in UIEB and SUIM) are computed, respectively. The CF-SI distribution is plotted in Figure 9. As presented, the distribution range of our selected UWIs is much broader than the other two datasets, which means that the features of our selected UWIs are more diverse. Moreover, it can also be found that some points are distributed closely in all three datasets, further indicating that there are many similar features between different UWIs.

Secondly, we also compare the performance of dictionaries trained with different data after embedding in the overall Dic-UCSNet. Note that only the training dataset for UF-Dic is different, while the training dataset for the CS network is the same. PSNR and SSIM are compared in UVTD-T at a sampling rate of 0.1 to show the performance. The experimental results are reported in Table 6. It is clearly seen that our dictionary can achieve better performance than the other two dictionaries. Combined with Figure 9, it can be found that the reconstruction performance is positively correlated with the distribution range of the CF-SI curve of the selected image. With the larger distribution of the CF-SI curve, the feature dictionary can provide richer reference features and thus improve the reconstruction quality.

#### A.5.2 EFFECT OF THE NUMBER OF CODEBOOKS ON THE RECONSTRUCTION QUALITY

As mentioned in Section 3.2, we build four scales codebooks $\mathbf{C_i}$ in which $i \in \{1, 2, 3, 4\}$ to store the features of UWIs. To verify the effect of different $i$ on performance, multiple comparison experiments are carried out on different $i$, where $i$ is set to $\in \{1, 2, 3, 4, 5, 6\}$ ($i = 1$ is the codebook with the maximum size of $128 \times 128$, and each increase in $i$ means adding a new codebook with half the

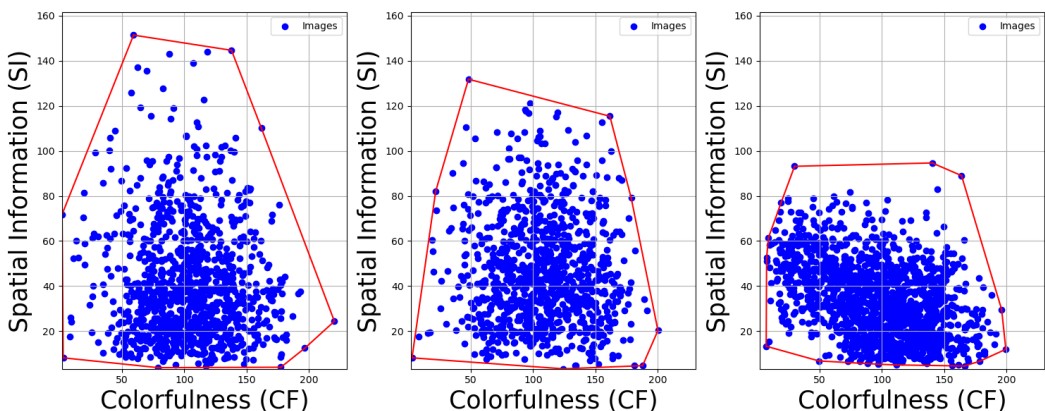

Figure 9: Comprehensiveness comparison of different UWIs for UF-Dic. The broader the CF-SI curve distributed, the more comprehensive the feature of UWIs are. From left to right is the CF-SI curve of UVTD-D, UIEB and SUIM

Table 6: Performance comparisons of different dictionary on UVTD dataset under the sampling rate of 0.1.

| Dictionary | PSNR↑ | SSIM↑ |
|---|---|---|
| UIEB | 30.13 | 0.887 |
| SUIM | 29.04 | 0.831 |
| Ours | **33.58** | **0.922** |

current size). We compare the PSNR and SSIM values of Dic-UCSNet in UVTD-T with different $i$ when the sampling rate is 0.1. The results are reported in Table 7. As shown, when $i$ increases from 1 to 4, the PSNR/SSIM value also increases rapidly. However, when $i \geq 4$, the improvement in the PSNR/SSIM value becomes slow, which means that the reference information that can be provided by continuing to add finer scale features becomes limited. Considering that increasing the number of codebooks will lead to a longer matching time of features and further increases the complexity of the entire framework, $i$ is set to four in our UF-Dic.

Table 7: Effect of the different number $i$ of codebooks.

| $i$ number | 1 | 2 | 3 | 4 | 5 | 6 |
|---|---|---|---|---|---|---|
| **PSNR↑** | 26.79 | 28.66 | 29.54 | 33.58 | 33.61 | 33.63 |
| **SSIM↑** | 0.861 | 0.882 | 0.898 | 0.922 | 0.925 | 0.928 |

### A.5.3 ROBUSTNESS AND GENERALIZATION ABILITY STUDY

In this section, some experiments are conducted to verify the robustness and generalizability of our proposed Dic-UCSNet.

**Robustness under environmental disturbances:** A robustness experiment is first conducted to prove the robustness of Dic-UCSNet under two external disturbances of the environment., turbidity and low lighting condition. First, three different Gaussian blurring kernels (with sizes 5, 7 and 9) are applied to the UWIs in UIEB to simulate the possible blurring interference. Secondly, the brightness of the UWIs in UIEB is reduced by 10%, 30% and 50% respectively to simulate low-light conditions. The PSNR/SSIM values of the reconstructed images are calculated from the original one. The experimental results are illustrated in Table 8. It can be found that, for blurred scenes, the performance of CSNet, ISTA-Net, and UCSNet declines significantly as the degree of blurring increases (with kernels 5, 7 and 9). The maximum drop reaches approximately 20 dB/0.700 on PSNR/SSIM compared to the results without distortions. However, our method experiences a

maximum decrease of only 9.64 dB/0.109 in PSNR/SSIM, which is a relatively acceptable range of decline. For low-light conditions, while our method shows a decrease of 10.23 dB/0.112 on PSN/SSIM when the brightness is reduced by 50%, this decline is still notably smaller than that of the other methods. We will take this as a key point for our future work.

Table 8: Robustness Comparisons on UIEB under the sampling rate of 0.1.

| Distortion Conditions | | CSNet$^+$ | | ISTA-Net$^+$ | | UCSNet | | Ours | |
|---|---|---|---|---|---|---|---|---|---|
| | | PSNR↑ | SSIM↑ | PSNR↑ | SSIM↑ | PSNR↑ | SSIM↑ | PSNR↑ | SSIM↑ |
| No distortions | | 28.62 | 0.816 | 28.48 | 0.806 | 33.57 | 0.902 | **34.24** | **0.913** |
| Gaussian Blur | kernel=5 | 22.10 | 0.550 | 21.95 | 0.535 | 27.36 | 0.821 | **31.25** | **0.882** |
| | kernel=7 | 16.07 | 0.408 | 15.24 | 0.387 | 20.38 | 0.522 | **27.94** | **0.848** |
| | kernel=9 | 9.58 | 0.119 | 9.21 | 0.108 | 14.02 | 0.318 | **24.60** | **0.804** |
| Low Light | -10% | 21.88 | 0.541 | 21.62 | 0.532 | 27.14 | 0.805 | **30.93** | **0.877** |
| | -30% | 15.14 | 0.345 | 14.63 | 0.339 | 20.43 | 0.535 | **27.51** | **0.839** |
| | -50% | 8.21 | 0.118 | 7.81 | 0.109 | 13.71 | 0.322 | **24.01** | **0.801** |

The **best** results are **bolded**.

**Generalization ability on arbitrary input resolution:** Since the size of training images for our Dic-UCSNet is 256×256, the generalization ability on arbitrary input resolution is important in terms of practical applications. An additional experiment where the input UWI size is the original size as in the dataset in the UIEB-challenge (Li et al., 2019) which contains 60 UWIs completely. As reported in Table 9, our method outperforms all other CS methods in both the quality of the reconstruction and the quality of the visual perception. Especially at extremely low sampling rates (0.01 and 0.04), our method can achieve at least a 10% performance improvement over the second-place method in all metrics. Furthermore, the improvement rate is also consistent across all sampling rates. Since the testing data are of any size, our method has a greater advantage over others in handling images of any size, demonstrating excellent robustness on input resolution.

Table 9: Generalization Ability Comparison on Arbitrary Input Resolution

| Sampling Rate | Metrics | AdapRecon | CSNet$^+$ | ISTA-Net$^+$ | DPC-DUN | TransCS | MTC-CSNet | UCSNet | Ours |
|---|---|---|---|---|---|---|---|---|---|
| 0.01 | PSNR↑ | 20.31 | 23.11 | 22.54 | 16.45 | 25.40 | 25.06 | 26.13 | **27.55** |
| | SSIM↑ | 0.575 | 0.704 | 0.633 | 0.411 | 0.729 | 0.714 | 0.764 | **0.777** |
| | LPIPS↓ | 0.653 | 0.562 | 0.671 | 0.793 | 0.485 | 0.498 | 0.396 | **0.360** |
| | NIQE↓ | 13.91 | 10.41 | 11.52 | 17.79 | 9.81 | 9.91 | 1.68 | **7.16** |
| 0.04 | PSNR↑ | 23.66 | 24.46 | 25.82 | 27.40 | 28.66 | 26.63 | 32.53 | **32.71** |
| | SSIM↑ | 0.694 | 0.730 | 0.735 | 0.817 | 0.862 | 0.783 | 0.878 | **0.886** |
| | LPIPS↓ | 0.798 | 0.772 | 0.690 | 0.318 | 0.309 | 0.383 | 0.243 | **0.237** |
| | NIQE↓ | 9.63 | 9.50 | 9.41 | 9.10 | 8.70 | 8.85 | 8.49 | **7.53** |
| 0.1 | PSNR↑ | 25.81 | 26.02 | 27.01 | 31.54 | 29.40 | 28.60 | 35.62 | **36.64** |
| | SSIM↑ | 0.730 | 0.801 | 0.865 | 0.891 | 0.874 | 0.870 | 0.925 | **0.937** |
| | LPIPS↓ | 0.723 | 0.707 | 0.668 | 0.266 | 0.287 | 0.316 | 0.157 | **0.149** |
| | NIQE↓ | 9.23 | 8.67 | 8.20 | 7.68 | 8.07 | 8.10 | 7.45 | **6.36** |
| 0.3 | PSNR↑ | 29.38 | 30.66 | 29.23 | 34.19 | 34.07 | 34.99 | 36.34 | **37.44** |
| | SSIM↑ | 0.898 | 0.904 | 0.891 | 0.950 | 0.947 | 0.943 | 0.962 | **0.977** |
| | LPIPS↓ | 0.282 | 0.237 | 0.251 | 0.140 | 0.149 | 0.147 | 0.082 | **0.067** |
| | NIQE↓ | 7.94 | 7.82 | 7.95 | 6.62 | 7.65 | 7.45 | 6.36 | **5.97** |

The **best** results are **bolded**.

## A.6 COMPLEXITY STUDY

In this section, the computation complexity comparison of the FLOPs, trainable parameters and inference time required for one single UWI of all compared methods are reported. Notably, all the experiments are conducted on the same RTX 4090. The results are reported in Table 10. As shown, compared to UCSNet which has the overall second place performance, our method has a lower complexity and can improve performance by at least 15% (shown in Table 1). Meanwhile, our method can complete the inference for a single image in 40 ms (approximately 25 frames), which is basically sufficient to meet the requirements of industrial detection. It can also be found that the complexity of our method is mainly brought by the dictionary part. After removing the UF-Dic, in terms of FLOPs and parameters, there are only 24.55G and 1.38M respectively. By simplifying the complexity of the UF-Dic, real-time performance can be further improved, which is also a part of our future work (mentioned in Section A.10).

Table 10: Complexity Comparison of Different CS methods on UVTD-T.

| Method | FLOPs (G) | Param. (M) | Inference Time (ms) |
|---|---|---|---|
| AdapRecon | 1.64 | 0.31 | 24 |
| CSNet$^+$ | 24.31 | 0.58 | 17 |
| ISTA-Net$^+$ | 30.93 | 0.42 | 18 |
| DPC-DUN | 138.12 | 1.63 | 20 |
| TransCS | 25.88 | 1.50 | 23 |
| MTC-CSNet | 21.57 | 0.87 | 25 |
| UCSNet | 70.51 | 3.98 | 87 |
| Ours | 63.62 | 5.88 | 40 |
| Our w/o UF-Dic | 24.55 | 1.38 | / |

### A.7 VISUALIZATION EXAMPLES OF UNDERWATER PHYSICAL PRIORS

In this section, some examples of underwater physical priors of different UWIs are provided. As shown in Figure 10, the background light can represent the color tone of the entire image, and the transmission map reflects the distance of the objects from the camera. It can be found that the clarity of UWIs is strongly related to the transmission map, which is reflected in the fact that the clarity of the texture in the scene decreases with the distance (in the transmission map, darker parts indicate closer to the shooting position). The background light and transmission map together constitute the unique degradation process of UWIs, and in this paper they are estimated from the network (Lin et al., 2020), which is designed strictly according to the mathematical formulation of underwater degradation (Eq. 2).

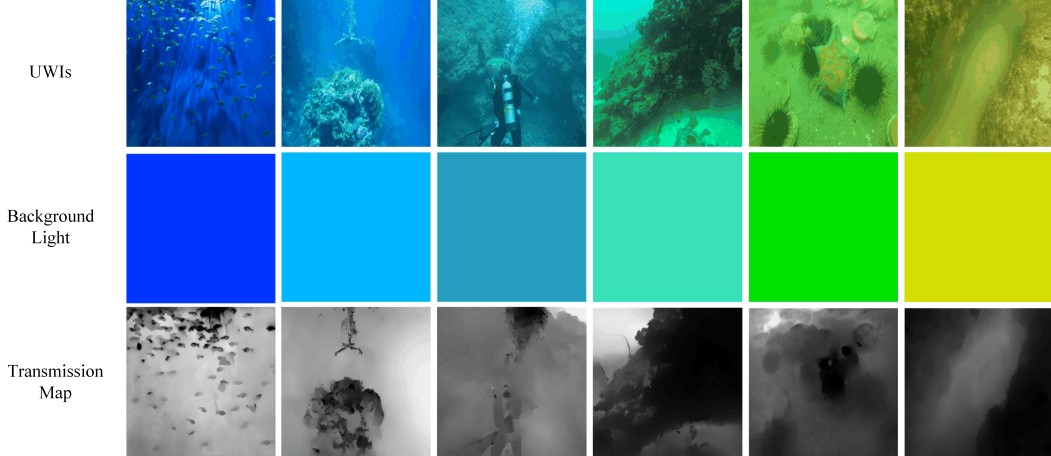

Figure 10: Examples for underwater physical priors of different UWIs under different water types.

### A.8 VISUALIZATION OF THE INTERMEDIATE FEATURES.

To demonstrate how the dictionary features are matched and used in our proposed Dic-UCSNet, the intermediate feature maps output from some key components of Dic-UCSNet are visualized in Figure 11 (the visualization results are feature maps in the reconstructed network at a sampling rate of 0.01). Note that we only visualized the first of the 64 feature channels for observation. Specifically, from the left to the right column is the original UWI, the feature map of the sampled results, the feature map of the reference features queried in the dictionary, the feature map of the reference features fined by DFVM, the feature map of the reference features fined by FMM and the output feature, respectively (corresponding to the data flow in Figure 4). As shown in Figure 11, the reference features quried from the UF-Dic have many similarities with the input features, which means that the input features can successfully match the reference features from the dictionary. Then after the deformation and adjustment by DFVM and FMM, input features with severe missing information can be well affined and the feature can become more abundant. These feature maps indicate that our

proposed Dic-UCSNet can successfully match and adjust reference features from the dictionary and utilize inter-UWIs similarity, which leads to good reconstruction quality at extremely low sampling rates.

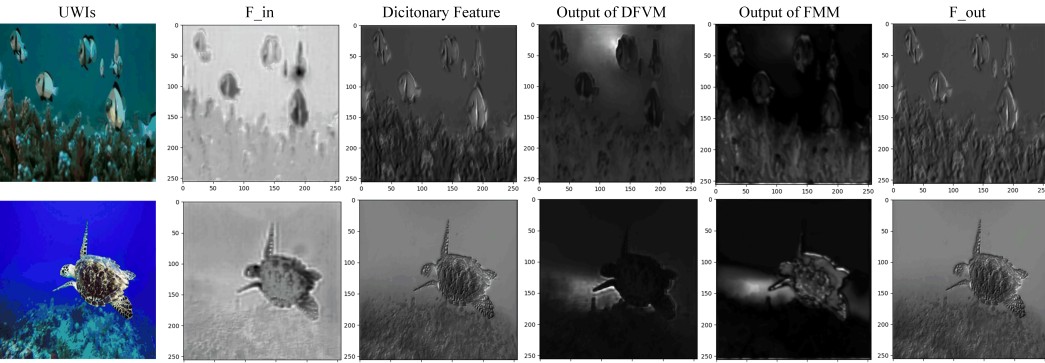

Figure 11: Visualization of the intermediate features in our Dic-UCSNet.

A.9 ADDITIONAL VISUAL EXPERIMENTAL RESULTS.

In this section, more visualization results of the qualitative comparison between our Dic-UCSNet and other SOTA CS methods are provided. Figure 12, Figure 13 and Figure 14 exhibit some examples from UVTD-T, UIEB and SUIM, respectively. Specifically, in each figure, from row 1 to row 3 is the result under the sampling rate of 0.01, 0.04 and 0.1. In addition, we also show the reconstructed images at two extremely low sampling rates (0.01 and 0.04) in Figure 15 and Figure 16, and more details are zoomed in for observation. From all these visualization results, it can be seen that at extremely low sampling rates, compared with other CS methods, the reconstruction results obtained by our Dic-UCSNet have better visual quality with richer details and sharper textures, and the objects in the scene can be easily distinguished by human eyes.

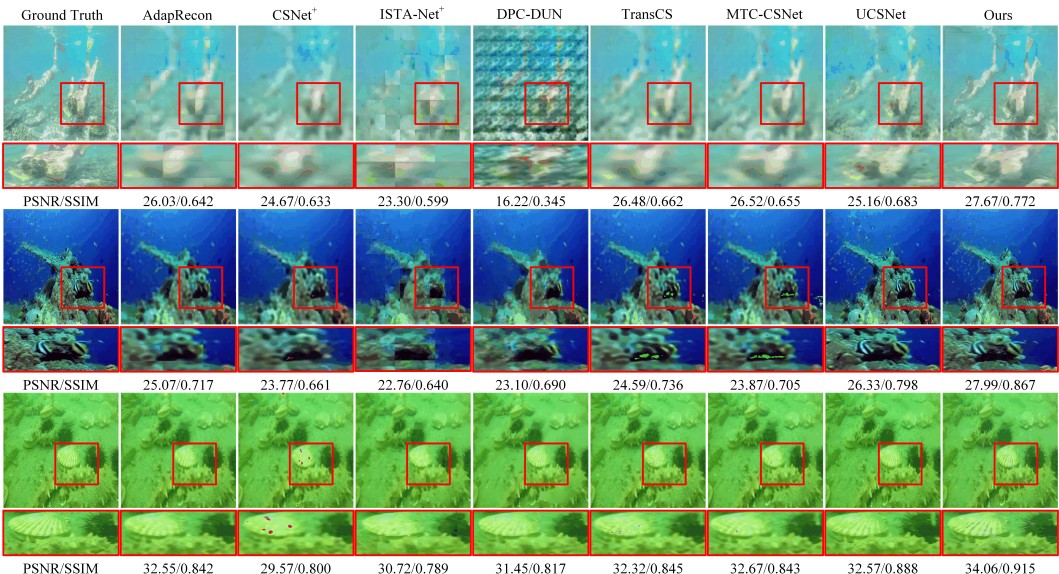

Figure 12: Visual quality comparisons on UVTD-T dataset. Row 1: examples at the sampling rate of 0.01; Row 2: examples at the sampling rate of 0.04; and Row 3: examples at the sampling rate of 0.1.

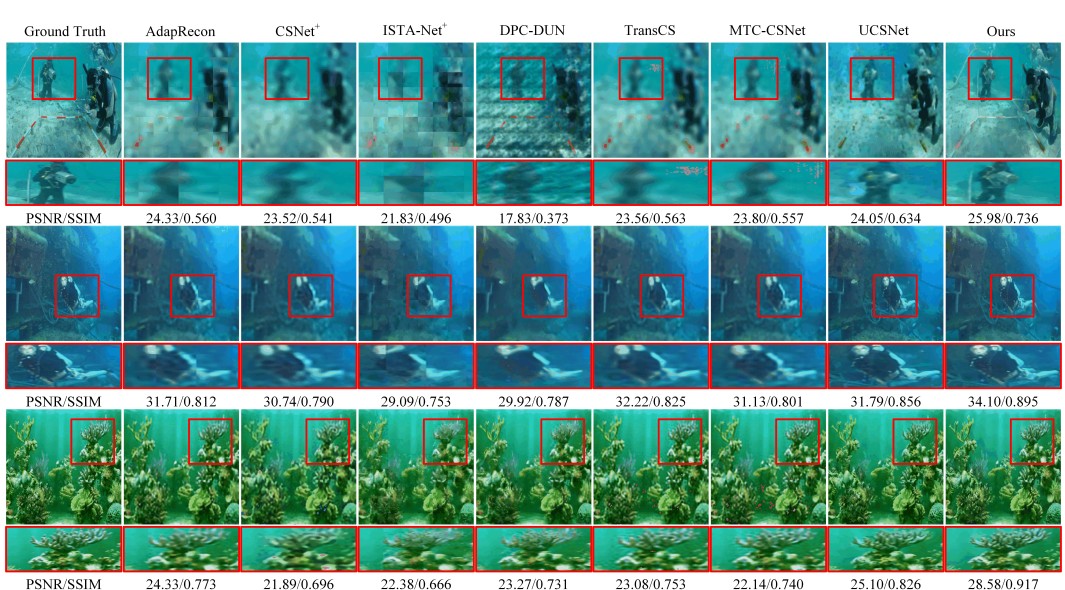

Figure 13: Visual quality comparisons on UIEB dataset. Row 1: examples at the sampling rate of 0.01; Row 2: examples at the sampling rate of 0.04; and Row 3: examples at the sampling rate of 0.1.

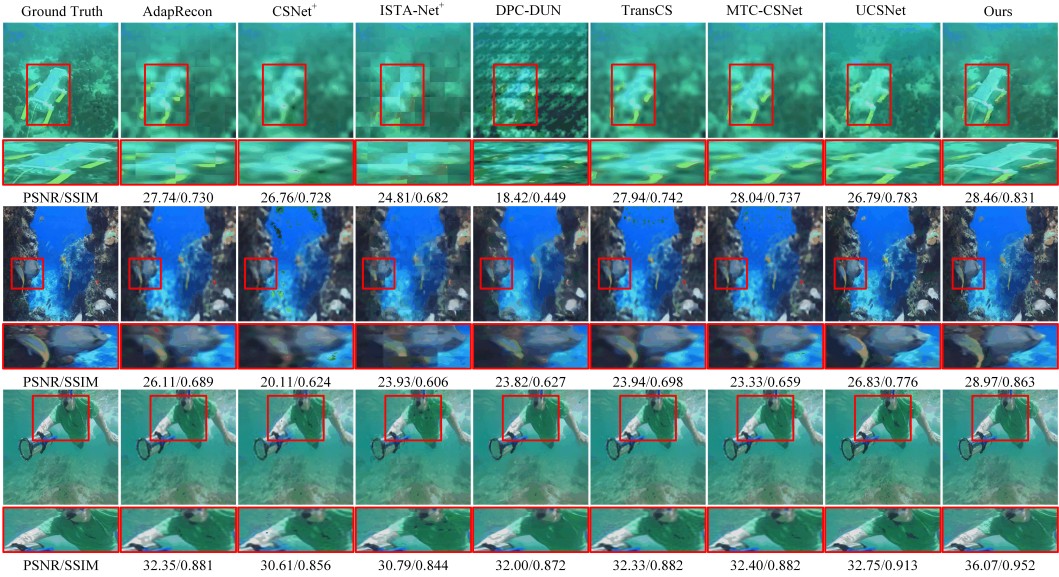

Figure 14: Visual quality comparisons on SUIM dataset. Row 1: examples at the sampling rate of 0.01; Row 2: examples at the sampling rate of 0.04; and Row 3: examples at the sampling rate of 0.1.

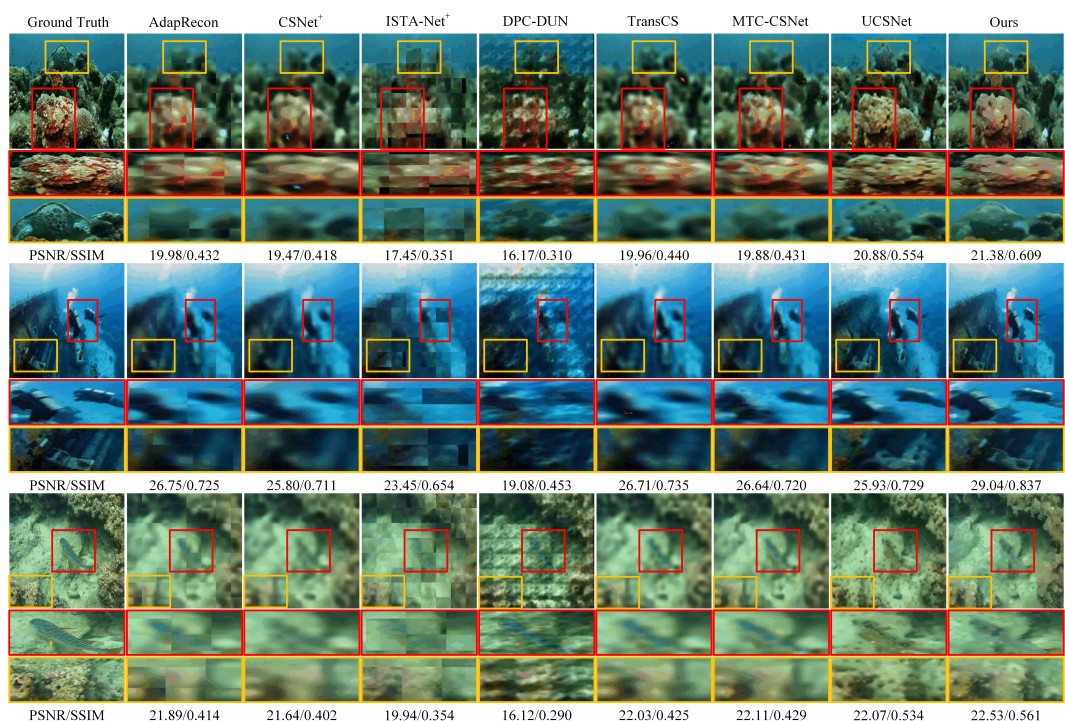

Figure 15: Visual quality comparisons at the sampling rate of 0.01. Row 1: examples from UVTD-T; Row 2: examples from UIEB; and Row 3: examples from SUIM.

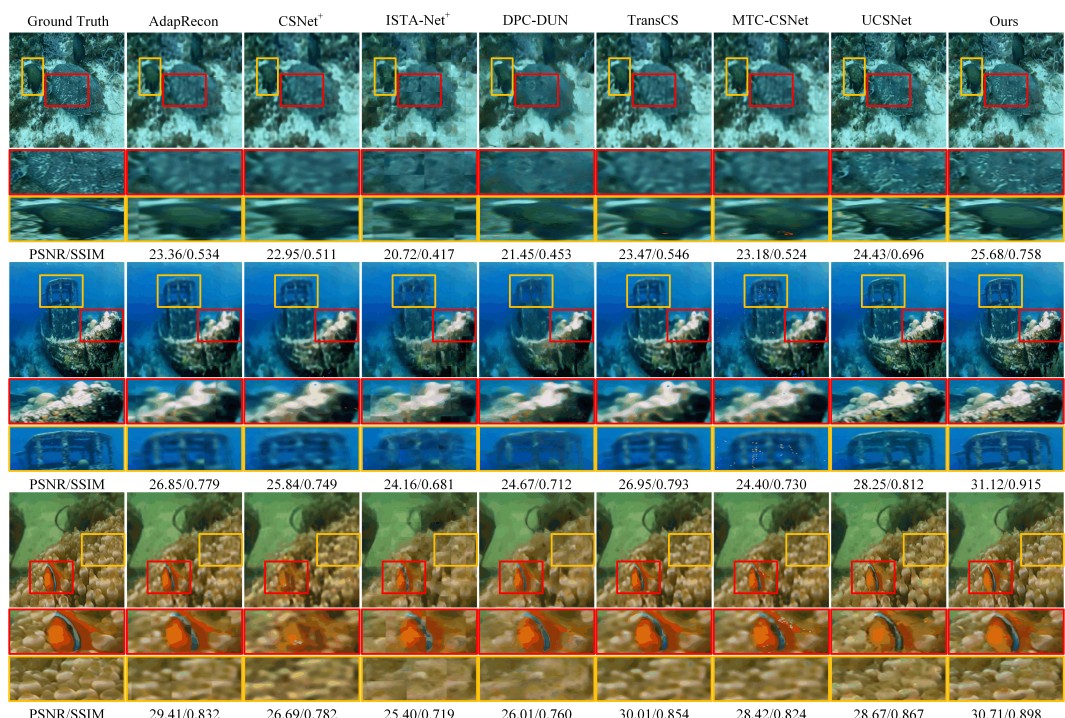

Figure 16: Visual quality comparisons at the sampling rate of 0.04. Row 1: examples from UVTD-T; Row 2: examples from UIEB; and Row 3: examples from SUIM.

### A.10   Limitations and future works

There are two main limitations to our proposed Dic-UCSNet. Firstly, our framework is composed of two parts, the ResBlock-based CS network and the GAN-based feature dictionary network. Meanwhile, in both the sampling and reconstruction processes, the dictionary needs to be queried. All of these lead to high complexity and parameter amount compared with other methods, which will have a certain impact on the real-time requirements in practical applications. Secondly, our Dic-UCSNet can only cope with a single sampling rate, and multiple networks need to be trained for different sampling rates. However, in practice, the sampling rate should change adaptively with the certain environment, including different bandwidth, different depth of water, etc. to meet the task requirements.

In future work, we will first study how to balance complexity with performance and how to reduce the complexity of underwater CS methods under the premise of ensuring performance. Second, we will study how to implement dynamic sampling in an underwater CS network to achieve multiple sampling rates in one model and improve deployability. Based on this, we will further study how to extend the image algorithm to the video domain.Finally, we will combine CS algorithm with coding technology to further reduce the data volume required for transmission or storage on the basis of sub-sampling, so as to achieve high-quality compression and transmission of UWIs at extremely low bit rates.

### A.11   Detailed architecture of our Dic-UCSNet.

The overall structure of our proposed Dic-UCSNet is provided in main paper in Figure 2. In this section, the detailed architectures and the detailed parameter setting of our sampling and reconstruction network, the key components MAM and FMM in UDFF-Module are provided in Table 11, Table 12, Table 13 and Table 14, respectively.

Table 11: Detailed structure of sampling network in our Dic-UCSNet.

| Input | UWIs |
|---|---|
| Layer 1 | Conv[3, 64, 3, 1, 1] + BatchNorm + ReLU |
| Layer 2 | ResidualBlock (He et al., 2016) |
| Layer 3 | ResidualBlock (He et al., 2016) |
| **Addition Input** | Reference feature + Underwater physical priors |
| Layer 4 | UDFF-Module(S) |
| **Addition Input** | Reference feature + Underwater physical priors |
| Layer 5 | UDFF-Module(S) |
| **Addition Input** | Reference feature + Underwater physical priors |
| Layer 6 | UDFF-Module(S) |
| **Addition Input** | Reference feature + Underwater physical priors |
| Layer 7 | UDFF-Module(S) |
| Layer 8 | Element Addition |
| Layer 9 | Matrix Generator (Zhuang et al., 2023) |
| **Output** | Sampling Matrices |

Conv[input channel, output channel, kernel size, stride, padding]
indicates the convolution layer.

Table 12: Detailed structure of reconstruction network in our Dic-UCSNet.

| Input | Sampled Image (UWIs × Sampling Matrices) |
| --- | --- |
| Layer 1 | Conv[3, 64, 3, 1, 1] + BatchNorm + ReLU |
| Layer 2 | ResidualBlock (He et al., 2016) |
| Layer 3 | ResidualBlock (He et al., 2016) |
| **Addition Input** | Reference feature + Underwater physical priors |
| Layer 4 | UDFF-Module(R) |
| **Addition Input** | Reference feature + Underwater physical priors |
| Layer 5 | UDFF-Module(R) |
| **Addition Input** | Reference feature + Underwater physical priors |
| Layer 6 | UDFF-Module(R) |
| **Addition Input** | Reference feature + Underwater physical priors |
| Layer 7 | UDFF-Module(R) |
| Layer 8 | Element Addition |
| Layer 9 | Conv[64, 3, 3, 1, 1] |
| **Output** | Reconstructed UWIs |

Conv[input channel, output channel, kernel size, stride, padding]
indicates the convolution layer.

Table 13: Detailed structure of MAM in UDFF-Module.

| Input | Image feature |
| --- | --- |
| Layer 1-1-1 | Conv[64, 64, 3, 1, 1] + ReLU
Conv[64, 64, 5, 1, 2] + ReLU
Conv[64, 64, 7, 1, 3] + ReLU |
| Layer 1-1-2 | Conv[192, 64, 1, 1, 0] + BatchNorm + ReLU |
| Layer 1-1-3 | Channel Concatenate |
| Layer 1-2-1 | ResidualBlock (He et al., 2016) |
| Layer 1-2-2 | ResidualBlock (He et al., 2016) |
| Layer 2 | Channel Concatenate |
| **Output** | Deconstructed image features |

Conv[input channel, output channel, kernel size, stride, padding]
indicates the convolution layer.

Table 14: Detailed structure of FMM in UDFF-Module.

| Input | Reference feature + Underwater physical priors |
| --- | --- |
| Layer 1 | DFVM transforms the reference features as shown in Figure 4. |
| **Output** | Normalized reference features |
| ⇓ | |
| **Input** | Deconstructed image features + Normalized reference features |
| Layer 1 | Channel Concatenate |
| Layer 2 | Conv[128, 64, 1, 1, 0] + ReLU + Conv[64, 64, 3, 1, 1] + Tanh |
| **Output** | Difference Map |
| ⇓ | |
| **Input** | Normalized reference features + Difference Map |
| Layer 1-1 | Conv[64, 64, 3, 1, 1] + ReLU |
| Layer 1-2 | Element Multiplication |
| Layer 2-1 | Conv[64, 64, 3, 1, 1] + ReLU |
| Layer 2-2 | Element Multiplication |
| Layer 3-1 | Conv[64, 64, 3, 1, 1] + ReLU |
| Layer 3-2 | Element Multiplication |
| **Output** | Fined Reference Feature |

Conv[input channel, output channel, kernel size, stride, padding]
indicates the convolution layer.

