# OpenReview forum: "Dic-UCSNet: A Novel Feature Dictionary-Based Underwater Image Compressive Sensing Framework"
_ICLR.cc/2026/Conference — ICLR 2026 Conference Withdrawn Submission_

### Official Review · Reviewer_Db5m · 2025-10-15

**Soundness:** 3
**Presentation:** 3
**Contribution:** 2
**Rating:** 4
**Confidence:** 4

**Summary:**

a feature dictionary-based CS framework for UWIs, dubbed Dic-UCSNet. Specifically, the authors first construct a multiscale discrete codebook as the underwater feature dictionary (UF-Dic), which can provide the inter-image similarity prior to underwater CS task. Subsequently, to better match the dictionary features with the input ones to improve the utilization of the dictionary features, they propose an underwater dictionary feature fusion module (UDFF-Module), which uses the underwater physical prior to transform the degradation style of the dictionary features to input ones, and then adaptively adjusts the dictionary features according to the difference map.

**Strengths:**

1. motivation
- The intuition that underwater scenes share repetitive textures and semantics due to environmental homogeneity is well-motivated and clearly demonstrated (e.g., Fig. 1 & Appendix A.1).

2. method
- The proposed UF-Dic (Underwater Feature Dictionary) built using multi-scale VQGAN codebooks is technically sound and scalable.
- The UDFF-Module (Underwater Dictionary Feature Fusion) effectively models degradation consistency using physical priors (background light & transmission map), improving realism and interpretability.

3. experiment
- comparisons on three real-world datasets (UVTD, UIEB, SUIM) demonstrate consistent gains across multiple metrics (PSNR, SSIM, LPIPS, NIQE), especially under extreme low sampling rates.
- Ablation studies are well designed.

**Weaknesses:**

1. novelty
- The core structure (sampling/reconstruction pipeline + underwater priors) is conceptually similar to UCSNet (Zhuang et al., 2023), which already incorporated underwater priors into CS.
- The paper might be seen as an extension of UCSNet with external feature injection, rather than a fundamentally new CS formulation.

2. cost
- Training VQGAN-based multi-scale dictionaries on large datasets (UVTD) is computationally heavy.
- The paper lacks discussion on storage size, lookup latency, and runtime efficiency of querying UF-Dic during inference — crucial for embedded or underwater systems with limited hardware.

3. Theory
- Although empirical results are strong, there is no theoretical or analytical discussion on why feature similarity improves compressive sensing recovery (e.g., sparsity relation, reconstruction bound).

4. Generalization
- The authors claim the method could be embedded in other CS frameworks, but all experiments are restricted to UWIs.
- Without at least one non-underwater validation, the claim of “generalizable dictionary-based CS” remains speculative.

**Questions:**

1. Efficiency
- How large is the multi-scale UF-Dic in terms of memory footprint?
- What is the average inference time compared to UCSNet or TransCS?
- Could the dictionary be compressed or partially loaded for resource-limited devices?

2. experiment
- The paper mentions that the UF-Dic is trained with selected 1000 UWIs, but the CS network is trained on separate 3504 images.
- How sensitive is performance to this split? Would reusing the same images for both stages lead to overfitting?
- Any experiments on transmission delay vs reconstruction quality trade-off?

3. generalization
- If a test image comes from a new underwater type or unseen semantic category, how well does the dictionary adapt?
- Is there a mechanism for incremental dictionary updating?

4. Comparison
- Were all baseline methods retrained on underwater data or used pretrained on natural images?
- Since transformer-based methods are sensitive to data domain, clarifying this would strengthen fairness.

---

### Official Review · Reviewer_wVuR · 2025-10-29

**Soundness:** 2
**Presentation:** 2
**Contribution:** 2
**Rating:** 4
**Confidence:** 3

**Summary:**

This paper proposes a novel feature dictionary-based framework for underwater image compressive sensing, named Dic-UCSNet. The authors observe that although underwater images suffer from severe degradation , a large number of similar features exist among different UWIs due to the relatively simple scenes and fewer semantics. By designing feature dictionary-based modules to leverage this inter-image similarity prior, the framework achieves significant performance improvements on the underwater CS task.

**Strengths:**

The paper's merits are confined to its conceptual starting point and its experimental breadth, rather than the credibility of its final results.

1. The conceptual premise is reasonable. The observation that UWIs possess inter-image similarity due to simpler semantics is valid. The attempt to formalize this domain-specific knowledge into an explicit, multi-scale feature dictionary (UF-Dic) is, in principle, an interesting direction for domain-specific CS.

2. The authors invested significant effort in the breadth of their empirical validation. For instance, the inclusion of robustness checks in Appendix A.5.3 (against Gaussian blur and low-light) and generalization tests (on arbitrary resolutions)  demonstrates commendable thoroughness. This holds even if the fairness of the primary SOTA comparison (Table 1) is fundamentally compromised by the methodological flaws discussed in the weaknesses.

**Weaknesses:**

Unfair parameter comparison and counterintuitive codebook prior.

The SOTA performance is likely an artifact of an unfair parameter comparison, not the efficacy of the dictionary architecture. The paper's methodology contains a fundamental flaw. To use the dictionary prior at the reconstruction end while avoiding the transmission of the codebook (which are extremely larger than the compressed data), the authors are forced to design the code predictor.

However, this predictor is tasked with a pathologically ill-posed problem: it must infer a codebook of 128*128+64*64+32*32+16*16=21,760 values from a tiny, heavily compressed signal of 655 values (at 0.01 SR if image is 256*256). In fact, the number of parameters in the proposed method is 5.88M, whereas for other methods, except for UCSNet (which has 3.98M), all have fewer than 2M parameters. More than that, in table 10, it’s lack of Our w/o UF-Dic’s inference time without any reason.

This invalidates the paper's central claim: that the SOTA performance stems from the "effectiveness of the dictionary prior." It is far more likely that the SOTA result is simply a product of the proposed model's (backbone + massive code predictor) vastly larger total parameter count compared to the baselines. The comparison in Table 1 is, therefore, highly unfair. The authors are comparing Baseline + massive predictor against Baseline.

**Questions:**

1. What would happen if the baseline parameters were the same as those in your method? The authors must provide a parameter-matched baseline to prove the true value of their method. For example, they must scale up a baseline until its total parameters equal the total parameters of the full. If this parameter-matched baseline also achieves SOTA, the paper's entire contribution regarding the "dictionary prior" is invalidated, as the gains would be shown to be a mere product of parameter brute-forcing.
2. What is Colorfulness and Spatial Information in Fig.9? More details are needed.

---

### Official Review · Reviewer_1FqY · 2025-10-29

**Soundness:** 2
**Presentation:** 3
**Contribution:** 2
**Rating:** 2
**Confidence:** 5

**Summary:**

The authors propose Dic-UCSNet, a feature dictionary-based compressed sensing framework for underwater images. It constructs a multi-scale underwater feature dictionary (UF-Dic) to provide inter-image similarity priors and introduces a dictionary feature fusion module (UDFF) that leverages physical priors to align and adaptively adjust dictionary features to the input. Results are reported on three real-world underwater image datasets.

**Strengths:**

1. The manuscript is well-organized, clearly written, and easy to follow.
2. According to the comparisons reported in the paper, the proposed method achieves superior performance over the baseline and competing approaches.

**Weaknesses:**

1. The main insight of the paper is that DifferentUWI softens semantically identical objects that share structural and feature similarities. However, this insight lacks sufficient supporting evidence and statistical analysis. Figure 1(a) is biased as it is based on a single case, especially considering that similar phenomena also exist in on-land images. Moreover, underwater environments contain diverse terrains and thousands of organisms, and the imaging conditions are complex, making it difficult to claim that underwater scenes exhibit more similar features compared to on-land scenes.
2. While the proposed framework is technically well-described, the network design is incremental and lacks theoretical depth. For instance, the combination of multi-scale features, dictionary learning, and the underwater imaging model does not offer sufficient novelty for a venue such as ICLR.

**Questions:**

Please see the Weaknesses.

---

### Official Review · Reviewer_m3yA · 2025-10-31

**Soundness:** 2
**Presentation:** 3
**Contribution:** 2
**Rating:** 4
**Confidence:** 3

**Summary:**

This paper proposes Dic-UCSNet, a feature dictionary-based compressive sensing framework specifically designed for underwater images. The key innovation lies in exploiting inter-image similarity among UWIs through a multi-scale VQGAN-based feature dictionary and an Underwater Dictionary Feature Fusion Module that adapts dictionary features using underwater physical priors. Experiments on three datasets show 5-15% improvements in reconstruction quality, particularly at extremely low sampling rates. The paper addresses a relevant problem, presents reasonable technical solutions, and reports promising experimental results. The application of feature dictionaries to underwater compressive sensing is novel, and the comprehensive evaluation demonstrates effectiveness. However, the limited algorithmic novelty, presentation issues, and missing statistical rigor prevent a higher rating. The work would benefit from addressing the computational complexity concerns and providing more robust statistical validation.

**Strengths:**

- Well-justified use of inter-UWI similarity with quantitative analysis.

- Testing on three datasets with multiple metrics and sampling rates.

- Tests under environmental disturbances and arbitrary resolutions.

**Weaknesses:**

- Although the paper proposes a “multi-scale feature dictionary” for cross-image feature reuse, similar ideas have already appeared in other vision domains such as face restoration, low-light enhancement, and codebook-based Transformer models.

- The framework assumes that different underwater images share similar structures and semantics. But in reality, changes in water type, depth, or lighting can drastically shift feature distributions. Since there’s no quantitative test on how performance drops with less similarity, it’s hard to tell how well the model generalizes.

- High parameter count (5.88M) and inference time (40ms) may limit practical deployment. It also requires separate training for different sampling rates, reducing practical flexibility.

- Missing error bars, significance tests, and multiple runs for robust evaluation. A dictionary trained only on UVTD may not generalize to diverse underwater conditions. No comparison with recent transformer-based CS methods like CSformer (2023) or SwinCS (2022).

**Questions:**

- How sensitive is the method to the dictionary size and composition? What happens with underwater scenes not represented in the training dictionary?

- The UF-Dic is trained offline on a fixed set of 1,000 UWI images (Sec. 3.2). Have the authors thought about letting the dictionary update dynamically or incrementally during inference? For example, when facing new water types or unseen objects, could it adapt without retraining from scratch? If not, please clarify whether a fixed dictionary might cause long-term performance decay.

- Table 1 shows strong gains at low sampling rates, but the paper doesn’t mention the extra compute or memory cost from UF-Dic querying, multi-scale matching, or the UDFF module. Could the authors share FLOPs, memory, or inference-time comparisons to confirm it’s still practical for real-time or low-power underwater systems like AUVs or ROVs?

- Can you provide statistical significance tests for the reported improvements, especially given the large performance gaps?

- How does the method perform with real-time varying sampling rates, which is crucial for practical underwater applications?

- What is the theoretical justification for the code predictor design? Have you considered more sophisticated prediction mechanisms?

---

### Official Review · Reviewer_7wL6 · 2025-11-10

**Soundness:** 3
**Presentation:** 3
**Contribution:** 3
**Rating:** 6
**Confidence:** 3

**Summary:**

This paper introduces Dic-UCSNet, a compressive sensing (CS) framework tailored for underwater images (UWIs). The key innovation is the use of a feature dictionary (UF-Dic) built from inter-UWI similarities to provide reference features during sampling and reconstruction. The framework includes a multi-scale VQGAN-based dictionary and an Underwater Dictionary Feature Fusion Module (UDFF-Module) that incorporates underwater physical priors for better feature matching. Experiments on real-world datasets (UVTD, UIEB, SUIM) demonstrate superior performance over state-of-the-art (SOTA) methods, particularly at low sampling rates (e.g., 0.01–0.04), with gains of 1–2 dB in PSNR and improvements in SSIM, LPIPS, and NIQE. Ablations validate the contributions of individual components, and the dictionary is shown to enhance other CS frameworks.

**Strengths:**

1. The paper is the first to exploit inter-UWI feature similarities via a dictionary for CS tasks. This is well-motivated by the analysis of UWI characteristics (e.g., simpler semantics leading to shared features across images), distinguishing it from land-based CS methods.
2. Quantitative comparisons on three datasets show consistent superiority, especially at extremely low rates where other methods fail. Visual results highlight better detail preservation and fewer artifacts.
3. The UDFF-Module effectively integrates physical priors, and ablations confirm its impact. Embedding UF-Dic into other frameworks yields significant PSNR gains, demonstrating plug-and-play potential.
4. Appendix covers motivations, ablations, robustness, and limitations, enhancing reproducibility.

**Weaknesses:**

1. As acknowledged in limitations (Appendix A.10), the framework has higher complexity (e.g., 40 ms inference vs. competitors in Table 5) due to dictionary queries and dual-network structure. No detailed FLOPs/param comparisons beyond Appendix A.6, and real-time applicability for underwater hardware is unclear.
2. The model requires retraining for each rate, limiting flexibility in dynamic environments (e.g., varying bandwidth). Adaptive sampling is suggested as future work but not addressed here.
3. Dictionary trained on ~1000 selected UVTD images; while diversity is shown (Figure 9), performance on non-optical UWIs (e.g., sonar) or extreme conditions (e.g., deep-sea) isn't tested. Robustness experiments (Table 8) are simulated, not real-world.
4. Misses some recent CS works (e.g., diffusion-based or hybrid models beyond cited ones).

**Questions:**

- How sensitive is performance to code predictor accuracy (Appendix A.2)? Provide error rates or ablations on prediction quality.
- In robustness tests (Table 8), why the minimal drop for Dic-UCSNet? Is it due to dictionary priors, and can you quantify this?
- UF-Dic selection uses greedy semantics (Appendix A.3); how does it perform on unbalanced datasets (e.g., rare objects)?
- Future work mentions video CS; any preliminary results on temporal extensions?
- Inference time breakdowns: How much is due to dictionary vs. CS network?

---

### Note · Authors · 2025-11-12

I have read and agree with the venue's withdrawal policy on behalf of myself and my co-authors.